# Deep neural networks to register and annotate cells in moving and deforming nervous systems

Adam A Atanas[1], Alicia Kun-Yang Lu[1], Brian Goodell[1], Jungsoo Kim[1], Saba N Baskoylu[1], Di Kang[1], Talya S Kramer[1], Eric Bueno[1], Flossie K Wan[1], Karen L Cunningham[2], Brandon Weissbourd[2], Steven W Flavell[1]*

[1]Picower Institute for Learning and Memory, Department of Brain and Cognitive Sciences, Howard Hughes Medical Institute, Massachusetts Institute of Technology, Cambridge, United States; [2]Picower Institute for Learning and Memory, Department of Biology, Massachusetts Institute of Technology, Cambridge, United States

## eLife Assessment

Whole-brain imaging of neuronal activity in freely behaving animals holds great promise for neuroscience, but numerous technical challenges limit its use. In this **important** study, the authors describe a new set of deep learning-based tools to track and identify the activity of head neurons in freely moving nematodes (*C. elegans*) and jellyfish (*Clytia hemisphaerica*). While the tools **convincingly** enable high tracking speed and accuracy in the settings in which the authors have evaluated them, the claim that these tools should be easily generalizable to a wide variety of datasets is incompletely supported.

*For correspondence:
flavell@mit.edu

**Abstract** Aligning and annotating the heterogeneous cell types that make up complex cellular tissues remains a major challenge in the analysis of biomedical imaging data. Here, we present a series of deep neural networks that allow for automatic non-rigid registration and cell identification, developed in the context of freely moving and deforming invertebrate nervous systems. A semi-supervised learning approach was used to train a *Caenorhabditis elegans* registration network (BrainAlignNet) that aligns pairs of images of the bending *C. elegans* head with single-pixel-level accuracy. When incorporated into an image analysis pipeline, this network can link neurons over time with 99.6% accuracy. This network could also be readily purposed to align neurons from the jellyfish *Clytia hemisphaerica*, an organism with a vastly different body plan and set of movements. A separate network (AutoCellLabeler) was trained to annotate >100 neuronal cell types in the *C. elegans* head based on multi-spectral fluorescence of genetic markers. This network labels >100 different cell types per animal with 98% accuracy, exceeding individual human labeler performance by aggregating knowledge across manually labeled datasets. Finally, we trained a third network (CellDiscoveryNet) to perform unsupervised discovery of >100 cell types in the *C. elegans* nervous system: by comparing multi-spectral imaging data from many animals, it can automatically identify and annotate cell types without using any human labels. The performance of CellDiscoveryNet matched that of trained human labelers. These tools should be immediately useful for a wide range of biological applications and should be straightforward to generalize to many other contexts requiring alignment and annotation of dense heterogeneous cell types in complex tissues.

## Introduction

Optical imaging of dense cellular tissues is widespread in biomedical research. Recently developed methods to label cells with highly multiplexed fluorescent probes should soon make it feasible to determine the heterogeneous cell types in any given sample (*Alon et al., 2021*; *Chen et al., 2015*; *Ke et al., 2013*). However, it remains challenging to extract critical information about cell identity and position from fluorescent imaging data. Aligning images within or across animals that have non-rigid deformations can be inefficient and lack cellular-level accuracy. Additionally, annotating cell types in a given sample can involve time-consuming manual labeling and often only results in coarse labeling of the main cell classes, rather than full annotation of the vast number of defined cellular subtypes.

Deep neural networks provide a promising avenue for aligning and annotating complex images of fluorescently labeled cells with high levels of efficiency and accuracy (*Moen et al., 2019*). Deep learning has generated high-performance tools for related problems, such as the general task of segmenting cells from background in images (*Stirling et al., 2021*; *Kirillov et al., 2023*). In addition, deep learning approaches have proven useful for non-rigid image registration in the context of medical image alignment (*Zou et al., 2022*) and for anatomical atlases (*Brezovec et al., 2024*). However, this has not been as widely applied to align images with single-cell resolution, which requires single-micron-level accuracy. Automated cell annotation using clustering of single-cell RNA sequencing data has been widely adopted (*Zidane et al., 2023*), but annotation of imaging data is more complex. Recent studies have shown the feasibility of using deep learning applied on image features (*Geuenich et al., 2021*) or raw imaging data to label major cell classes (*Zidane et al., 2023*; *Amitay et al., 2023*; *Brbić et al., 2022*). However, these methods are still not sufficiently advanced to label the potentially hundreds of cellular subtypes in images of complex tissues. In addition, fully unsupervised discovery of the many distinct cell types in cellular imaging data remains an unsolved challenge.

There is considerable interest in using these methods to automatically align and annotate cells in the nervous system of *Caenorhabditis elegans*, which consists of 302 uniquely identifiable neurons (*White et al., 1986*; *Witvliet et al., 2021*; *Cook et al., 2019*). The optical transparency of the animal enables in vivo imaging of fluorescent indicators of neural activity at brain-wide scale (*Prevedel et al., 2014*; *Schrödel et al., 2013*). Advances in closed-loop tracking made this imaging feasible in freely moving animals (*Nguyen et al., 2016*; *Venkatachalam et al., 2016*). These approaches are being used to map the relationship between brain-wide activity and flexible behavior (reviewed in *Flavell and Gordus, 2022*; *Kramer and Flavell, 2024*). However, analysis is still impeded by how the animal bends and warps its head as it moves, resulting in non-rigid deformations of the densely packed cells in its nervous system. Fully automating the alignment and annotation of cells in *C. elegans* imaging data would facilitate high-throughput and high-SNR brain-wide calcium imaging. These methods would also aid studies of reporter gene expression, developmental trajectories, and more. Moreover, similar challenges exist in a variety of other organisms whose nervous systems undergo deformations, like hydra, jellyfish, *Drosophila* larvae, and others.

Previous studies have described methods to align and annotate cells in multicellular imaging datasets from *C. elegans* and hydra. Datasets from freely moving animals pose an especially challenging case. Methods for aligning cells across timepoints in moving datasets include approaches that link neurons across adjacent timepoints (*Lagache et al., 2021*; *Hanson et al., 2024*; *Wen et al., 2021*), as well as approaches that use signal demixing (*Nejatbakhsh et al., 2020*), alignment of body position markers using anatomical constraints (*Christensen et al., 2015*; *Ardiel et al., 2022*), or registration/clustering/matching based on features of the neurons, such as their centroid positions (*Ryu et al., 2024*; *Nguyen et al., 2017*; *Yu et al., 2021*; *Wu et al., 2022*; *Nejatbakhsh and Varol, 2021*; *Deng et al., 2024*; *Fieseler et al., 2025*). Targeted data augmentation combined with deep learning applied to raw images has recently been used to reduce manual labeling time during cell alignment (*Park et al., 2024*). Deep learning applied to raw images has also been used to identify specific image features, like multicellular structures in *C. elegans* (*Wang et al., 2019*). We have previously applied non-rigid registration to full fluorescent images from brain-wide calcium imaging datasets to perform neuron alignment, but performing this complex image alignment via gradient descent is very slow, taking multiple days to process a single animal's data even on a computing cluster (*Atanas et al., 2023*). In summary, all these current methods for neuron alignment are constrained by a tradeoff between alignment accuracy and processing time, either due to manually labeling subsets of neurons or computing the complex alignments required to yield >95% alignment accuracy.

For *C. elegans* neuron class annotation, ground-truth measurements of neurons' locations in the head have allowed researchers to develop atlases describing the statistical likelihood of finding a given neuron in a given location (*Toyoshima et al., 2020*; *Bubnis et al., 2019*; *Chaudhary et al., 2021*; *Skuhersky et al., 2022*; *Yemini et al., 2021*; *Varol et al., 2020*; *Sprague et al., 2024*). Some of these atlases used the NeuroPAL transgene in which four fluorescent proteins are expressed in genetically defined sets of cells, allowing users to manually determine their identity based on multi-spectral fluorescence and neuron position (*Yemini et al., 2021*; *Varol et al., 2020*; *Sprague et al., 2024*). However, this manual labeling is time-consuming (hours per dataset), and statistical approaches to automate neuron annotation based on manual labeling have still not achieved human-level performance (>95% accuracy).

Here, we describe deep neural networks that solve these alignment and annotation tasks. First, we trained a neural network (BrainAlignNet) that can perform non-rigid registration to align images of the worm's head from different timepoints in freely moving data. It is >600 fold faster than our previous gradient descent-based approach using elastix (*Atanas et al., 2023*) and aligns neurons with 99.6% accuracy. Moreover, we show that this same network can be easily purposed to align cells recorded from the jellyfish *C. hemisphaerica*, an organism with a vastly different body plan and pattern of movement. Second, we trained a neural network (AutoCellLabeler) that annotates the identity of each *C. elegans* head neuron based on multi-spectral NeuroPAL labels, which achieves 98% accuracy. Finally, we trained a different network (CellDiscoveryNet) that can perform unsupervised discovery and labeling of >100 cell types of the *C. elegans* nervous system with 93% accuracy by comparing unlabeled NeuroPAL images across animals. Overall, our results reveal how to train neural networks to automatically identify and annotate cells in complex cellular imaging data with high performance. These tools will be immediately useful for many biological applications and can be easily extended to solve related image analysis problems in other systems.

## Results

### BrainAlignNet: a neural network that registers cells in the deforming head of freely moving *C. elegans*

When analyzing neuronal calcium imaging data, it is essential to accurately link neurons' identities over time to construct reliable calcium traces. This task is challenging in freely moving animals where animal movement warps the nervous system on sub-second timescales. Therefore, we sought to develop a fast and accurate method to perform non-rigid image registration that can deal with these warping images. Previous studies have described such methods for non-rigid registration of point clouds (e.g. neuron centroid positions; *Nguyen et al., 2017*; *Yu et al., 2021*; *Wu et al., 2022*; *Ma et al., 2016*), but, as we describe below, we found that performing full image alignment allows for higher accuracy neuron alignment.

To solve this task, we used a previously described network architecture (*Fu et al., 2020*; *Hu et al., 2018*) that takes as input a pair of 3-D images (i.e. volumes of fluorescent imaging data of the head of the worm) from different timepoints of the same neural recording (*Figure 1A*). The network is tasked with determining how to warp one 3-D image (termed the 'moving image') so that it resembles the other 3-D image (termed the 'fixed image'). Specifically, the network outputs a dense displacement field (DDF), a pixel-wise coordinate transformation function designed to indicate which points in the moving and fixed images are the same (see Materials and methods). The moving image is then transformed through this DDF to create a warped moving image, which should look like the fixed image. This network was selected because its LocalNet architecture (a modified 3-D U-Net) allows it to do the feature extraction and image reconstruction necessary to solve the task. To train and evaluate the network, we used data from freely moving animals expressing both pan-neuronal NLS-GCaMP and NLS-tagRFP, but only provided the tagRFP images to the network, since this fluorophore's brightness should remain static over time. Since Euler registration of images (rotation and translation) is simple, we performed Euler registration on the images using a GPU-accelerated grid search prior to inputting them into the network. During training, we also provided the network with the locations of the centroids of matched neurons found in both images, which were available for these training and validation data since we had previously used gradient descent to solve those registration problems ('registration problem' here is defined as a single image pair that needs to be aligned) and link

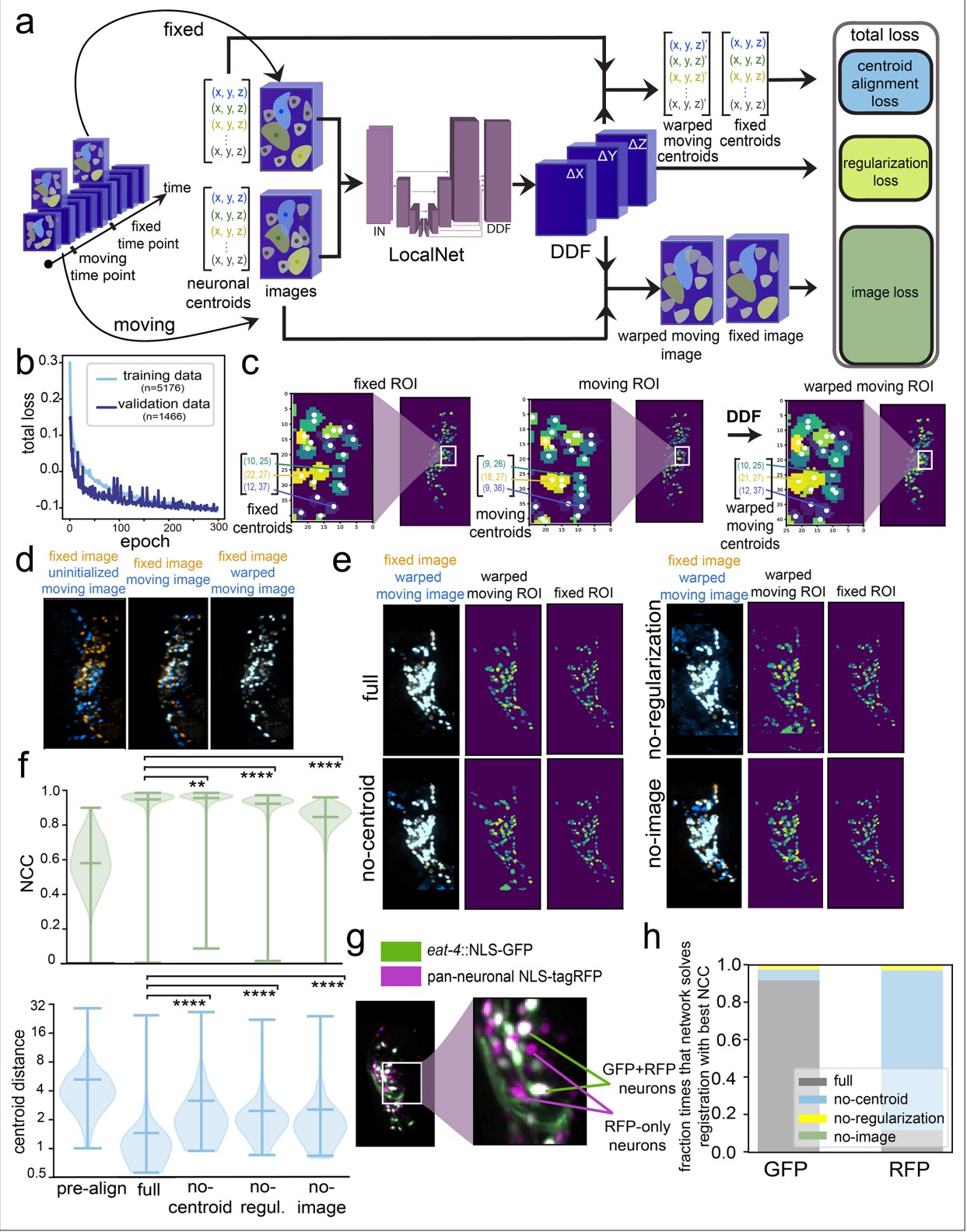

**Figure 1.** BrainAlignNet can perform non-rigid registration to align the neurons in the *C. elegans* head. (**A**) Network training pipeline. The network takes in a pair of images and a pair of centroid position lists corresponding to the images at two different time points (fixed and moving). (In the LocalNet diagram, this is represented as 'IN'. Intermediate cuboids represent intermediate representations of the images at various stages of network processing. In reality, the cuboids are four-dimensional, but we represent them with three dimensions (up/down is *x*, left/right is *y*, in/out is channel, and we omit *z*) for visualization purposes. Spaces and arrows between cuboids represent network blocks, layers, and information flow. See Materials and

*Figure 1 continued on next page*

*Figure 1 continued*

methods for a detailed description of network architectures.) Image pairs were selected based on the similarity of worm postures (see Materials and methods). The fixed and moving images were pre-registered using an Euler transformation, translating and rotating the moving images to maximize their cross-correlation with the fixed images. The fixed and moving neuron centroid positions were obtained by computing the centers of the same neurons in both the fixed and moving images as a list of (x, y, z) coordinates. This information was available since we had previously extracted calcium traces from these videos using a previous, slow version of our image analysis pipeline. The network outputs a Dense Displacement Field (DDF), a 4-D tensor that indicates a coordinate transformation from fixed image coordinates to moving image coordinates. The DDF is then used to transform the moving images and fixed centroids to resemble the fixed images and moving centroids. During training, the network is tasked with learning a DDF that transforms the centroids and images in a way that minimizes the centroid alignment and image loss, as well as the regularization loss (see Materials and methods). Note that, after training, only images (not centroids) need to be input into the network to align the images. (**B**) Network loss curves. The training and validation loss curves show that validation performance plateaued around 300 epochs of training. (**C**) Example of registration outcomes on neuronal ROI images. The network-learned DDF warps the neurons in the moving image ('moving ROIs'). The warped-moving ROIs are meant to be closer to the fixed ROIs. Each neuron is uniquely colored in the ROI images to represent its identity. The centroids of these neurons are represented by the white dots. Here, we take a z-slice of the 3-D fixed and moving ROI blocks on the x-y plane to show that the DDF can warp the x and y coordinates of the moving centroids to align with the x and y coordinates of the fixed centroids with one-pixel precision. (**D**) Example of registration outcomes on tagRFP images. We show the indicated image blocks as Maximal Intensity Projections (MIPs) along the z-axis, overlaying the fixed image (orange) with different versions of the moving image (blue). While the fixed image remains untransformed, the uninitialized moving image (left) gets warped by an Euler transformation (middle) and a network-learned DDF (right) to overlap with the fixed image. (**E**) Registration outcomes shown on example tagRFP and ROI images for four different trained networks. We randomly selected one registration problem from one of the testing datasets and tasked the trained networks with creating a DDF to warp the moving (RFP) image and moving ROI onto the fixed (RFP) image and fixed ROI. The full network with full loss function aligns neurons in both RFP and ROI images almost perfectly. For the networks trained without the centroid alignment loss, regularization loss, or image loss—while keeping the rest of the training configurations identical—the resulting DDF is unable to fully align the neurons and displays unrealistic deformation (closely inspect the warped moving ROI images). (**F**) Evaluation of registration performance on testing datasets before network registration and after registration with four different networks. 'pre-align' shows alignment statistics on images after Euler alignment, but before neural network registration. Two performance metrics are shown. Normalized cross-correlation (NCC, top) quantifies alignment of the fixed and warped moving RFP images, where a score of one indicates perfect alignment. Centroid distance (bottom) is measured as the mean Euclidean distance between the centroids of all neurons in the fixed ROI and the centroids of their corresponding neurons in the warped moving ROI; a distance of 0 indicates perfect alignment. All violin plots are accompanied by lines indicating the minimum, mean, and maximum values. \*\*p<0.01, \*\*\*p<0.001, \*\*\*\*p<0.0001, distributions of registration metrics (NCC and centroid distance) were compared pairwise across all four versions of the network with the two-tailed Wilcoxon signed rank test *only* on problems that register frames from unique timepoints. For all networks ('pre-align', 'full', 'no-centroid', 'no-regul.', 'no-image'), we evaluated performance on the same registration problems drawn from five animals in the testing set (n=447 total registration problems from those five animals). For NCC, W=35,281, 64,854, 78,754 for 'no-centroid', 'no-regul.', 'no-image' vs 'full', respectively; for centroid distance, W=12,168, 12,634, 13,345 for 'no-centroid', 'no-regul.', 'no-image' vs. 'full', respectively. (**G**) Example image of the head of an animal from a strain that expresses both pan-neuronal NLS-tagRFP and eat-4::NLS-GFP. The neurons expressing both NLS-tagRFP and eat-4::NLS-GFP are a subset of all the neurons expressing pan-neuronal NLS-tagRFP. (**H**) A comparison of the registration qualities of the four trained registration networks: full network, no-centroid alignment loss, no-regularization loss, no-image loss. Each network was evaluated on four datasets in which both pan-neuronal NLS-tagRFP and *eat-4*::NLS-GFP are expressed, examining 3927 registration problems per dataset. For a total of 15,708 registration problems, each network was tasked with registering the tagRFP images. The resulting DDFs from the tagRFP registrations were also used to register the *eat-4*::GFP images. For each channel in each problem, we determined which of the four networks had the highest performance (i.e. highest NCC). Note that the no-centroid alignment network performs the best of the RFP channel, but not in the GFP channel. Instead, the full network performs the best in the GFP channel. This suggests that the network without the centroid alignment loss deforms RFP images in a manner that does not accurately move the neurons to their correct locations (i.e. scrambles the pixels).

The online version of this article includes the following figure supplement(s) for figure 1:

**Figure supplement 1.** Example images and performance of the network trained to register arbitrary image pairs.

neurons' identities (*Atanas et al., 2023*). The centroid locations are only used for network training and are not required for the network to solve registration problems after training. The loss function that the network was tasked with minimizing had three components: (1) image loss: negative of the Local squared zero-Normalized Cross-Correlation (LNCC) of the fixed and warped moving RFP images, which takes on a higher value when the images are more similar (hence making the image loss more negative); (2) centroid alignment loss: the average of the Euclidean distances between the matched centroid pairs, where lower values indicate better alignment; and (3) regularization loss: a term that increases the overall loss the more that the images are deformed in a non-rigid manner (in particular, penalizing image scaling and scrambling of adjacent pixels; see Materials and methods).

We trained and validated the network on 5176 and 1466 image pairs (from 57 and 22 animals), respectively, over 300 epochs (*Figure 1B*). We then evaluated network performance on a separate set of 447 image pairs reserved for testing that were recorded from different animals. On average, the network improved the Normalized Cross-Correlation (NCC; an overall metric of image similarity,

where 1.0 means images are identical; see Materials and methods for equation) from 0.577 in the input image pairs to 0.947 in the registered image pairs (*Figure 1C* shows example of centroid positions; *Figure 1D* shows image example; *Figure 1E* shows both; *Figure 1F* shows quantification of pre-aligned and registered). After alignment, the average distance between aligned centroids was 1.45 pixels (*Figure 1F*). These results were only modestly different depending on the animal or the exact registration problem being solved (*Figure 1—figure supplement 1A–C*).

To determine which features of the network were critical for its performance, we trained three additional networks, using loss functions where we omitted either the centroid alignment loss, the regularization loss, or the image loss. In the first case, the network would not be able to learn based on whether the neuron centroids were well-aligned; in the second case, there would be no constraints on the network performing any type of deformation to solve the task; in the third case, the deformations that the network learned to apply could only be learned from the alignment of the centroids, not the raw tagRFP images. Registration performance of each network was evaluated using the NCC and centroid distance, which quantify the quality of tagRFP image alignment and centroid alignment, respectively (*Figure 1F*). While the NCC scores were similar for the full network and the no-regularization and no-centroid alignment networks, other performance metrics like centroid distance were significantly impaired by the absence of centroid alignment loss or regularization loss (*Figure 1E–F*). This suggests that in the absence of centroid alignment loss or regularization loss, the network learns how to align the tagRFP images, but does so using unnatural deformations that do not reflect how the worm bends. In the case of the no-image loss network, all performance metrics, including both image and centroid alignment, were impaired compared to the full network (*Figure 1F*). This suggests that requiring the network to learn how to warp the RFP images enhances the network's ability to learn how to align the neuron positions (i.e. centroids). Thus, the loss on RFP image alignment can serve to regularize point cloud alignment.

The finding that the centroid positions were precisely aligned by the full network indicates that the centers of the neurons were correctly registered by the network. However, it does not ensure that all of the pixels that comprise a neuron are correctly registered, which could be important for subsequent feature extraction from the aligned images. For example, it is formally possible to have perfect RFP image alignment in a context where the pixels from one neuron in the moving RFP image are scrambled to multiple neuron locations in the warped moving RFP image. In fact, we observed this in our first efforts to build such a network, where the loss function was only composed of the image loss. As an additional control to test for this possibility in our trained networks, we examined the network's performance on data from a different strain that expresses pan-neuronal NLS-mNeptune (analogous to the pan-neuronal NLS-tagRFP) and *eat-4*::NLS-GFP, which is expressed in ~40% of the neurons in the *C. elegans* head (*Figure 1G* shows example image). If the pixels within the neurons are being correctly registered, then applying image registration to the GFP channel for these image pairs should result in highly correlated images (i.e. a high NCC value close to 1). If the pixels within neurons are being scrambled, then these images should not be well-aligned. This test is somewhat analogous to building a ground-truth labeled dataset, but does not require manual labeling and is less susceptible to human labeling errors. We used the DDF that the network output based on the pan-neuronal mNeptune data to register the corresponding *eat-4*::NLS-GFP images from the same timepoints and found that this resulted in high-quality GFP image alignment (*Figure 1H*). In contrast, while the no-centroid alignment and no-regularization networks output a DDF that successfully aligned the RFP images, applying this DDF to corresponding GFP images resulted in poor GFP image registration (*Figure 1H* shows that the no-centroid alignment network aligns the RFP channel, but not the GFP channel, in the *eat-4*::NLS-GFP strain). This further suggests that these reduced networks lacking centroid alignment or regularization loss are aligning the RFP images through unnatural image deformations. Together, these results suggest that the full **Brain Align**ment Neural **Net**work (**BrainAlignNet**) can perform non-rigid registration on pairs of images from freely moving animals.

The registration problems included in the training, validation, and test data above were pulled from a set of registration problems that we had been able to solve with gradient descent (example images in *Figure 1—figure supplement 1D*). These problems did not include the most challenging cases, for example, when the two images to be registered had the worm's head bent in opposite directions (although we note that they did include substantial non-rigid deformations). We next asked whether a network trained on arbitrary registration problems, including those that were not solvable with

gradient descent (example images in *Figure 1—figure supplement 1E*), could obtain high performance. For this test, we also omitted the Euler registration step that we performed in advance of network training, since the goal was to test whether this network architecture could solve any arbitrary *C. elegans* head alignment problem. For this analysis, we used the same loss function as the successful network described above. We also increased the amount of training data from 5176 to 335,588 registration problems. The network was trained for 300 epochs. However, the test performance of the network was not high in terms of image alignment or centroid alignment (*Figure 1—figure supplement 1F*). This suggests that additional approaches may be necessary to solve these more challenging registration problems. Overall, our results suggest that, provided there is an appropriate loss function, a deep neural network can perform non-rigid registration problems to align neurons across the *C. elegans* head with high speed and single-pixel-level accuracy.

## Integration of BrainAlignNet into a complete calcium imaging processing pipeline

The above results suggest that BrainAlignNet can perform high-quality image alignments. The main value of performing these alignments is to enable accurate linking of neurons over time. To test whether performance was sufficient for this, we incorporated BrainAlignNet into our existing image analysis pipeline for brain-wide calcium imaging data and compared the results to our previously described pipeline, which used gradient descent to solve image registration (*Atanas et al., 2023*). This image analysis pipeline, the **A**utomated **N**euron **T**racking **S**ystem for **U**nconstrained **N**ematodes (ANTSUN), includes steps for neuron segmentation (via a 3D U-Net), image registration, and linking of neurons' identities (*Figure 2A*). Several steps are required to link neurons' identities based on registration of these high-complexity 3D images. First, image registration defines a coordinate transformation between the two images, which is then applied to the segmented neuron ROIs, warping them into a common coordinate frame. To link neuron identities over time, we then build an N-by-N matrix (where N is the number of all segmented neuron ROIs at all timepoints in a given recording, i.e. approximately # ROIs * # timepoints) with the following structure: (1) Enter zero if the ROIs were in an image pair that was not registered (we do not attempt to solve all registration problems, as this is unnecessary); (2) Enter zero if the ROIs were from a registered image pair, but the registration-warped ROI did not overlap with the fixed ROI; and (3) Otherwise, enter a heuristic value indicating the confidence that the ROIs are the same neurons based on several ROI features. These features include similarity of ROI positions and sizes, similarity of red channel brightness, registration quality, a penalty for overly nonlinear registration transformations, and a penalty if ROIs were displaced over large distances during alignment. Finally, custom hierarchical clustering is applied to the matrix to generate clusters consisting of the ROIs that reflect the same neuron recorded at different time points. Calcium traces are then constructed from all of these timepoints, normalizing the GCaMP signal to the tagRFP signal. We term the ANTSUN pipeline with gradient descent registration ANTSUN 1.4 (*Atanas et al., 2023*; *Pradhan et al., 2025*) and the version with BrainAlignNet registration ANTSUN 2.0 (*Figure 2A*).

We ran a series of control datasets through both versions of ANTSUN to benchmark their results. The first was from animals with pan-neuronal NLS-mNeptune and *eat-4*::NLS-GFP, described above. The resulting GFP traces from these recordings allow us to quantify the number of timepoints where the neuron identities are not accurately linked together into a single trace (*Figure 2B* shows example dataset). Specifically, in this strain, this type of error can be easily detected since it can result in a low-intensity GFP neuron (*eat-4*-) suddenly having a high-intensity value when the trace mistakenly incorporates data from a high-intensity neuron (*eat-4*+), or vice versa. We computed this error rate, taking into account the overall similarity of GFP intensities (i.e. since we can only observe errors when GFP- and GFP+ neurons are combined into the same trace). For both versions of ANTSUN, the error rates were <0.5%, suggesting that >99.5% of timepoints reflect correctly linked neurons (*Figure 2C*).

We next estimated motion artifacts in the data collected from ANTSUN 2.0, as compared to ANTSUN 1.4. Here, we processed data from three pan-neuronal GFP datasets. GFP traces should ideally be constant, so the fluctuations in these traces provide an indication of any possible artifacts in data collection or processing (*Figure 2—figure supplement 1A* shows example dataset). Quantification of signal variation in GFP traces showed similar results for ANTSUN 1.4 and 2.0, suggesting that incorporating BrainAlignNet did not introduce artifacts that impair signal quality of the traces (*Figure 2—figure supplement 1B*).

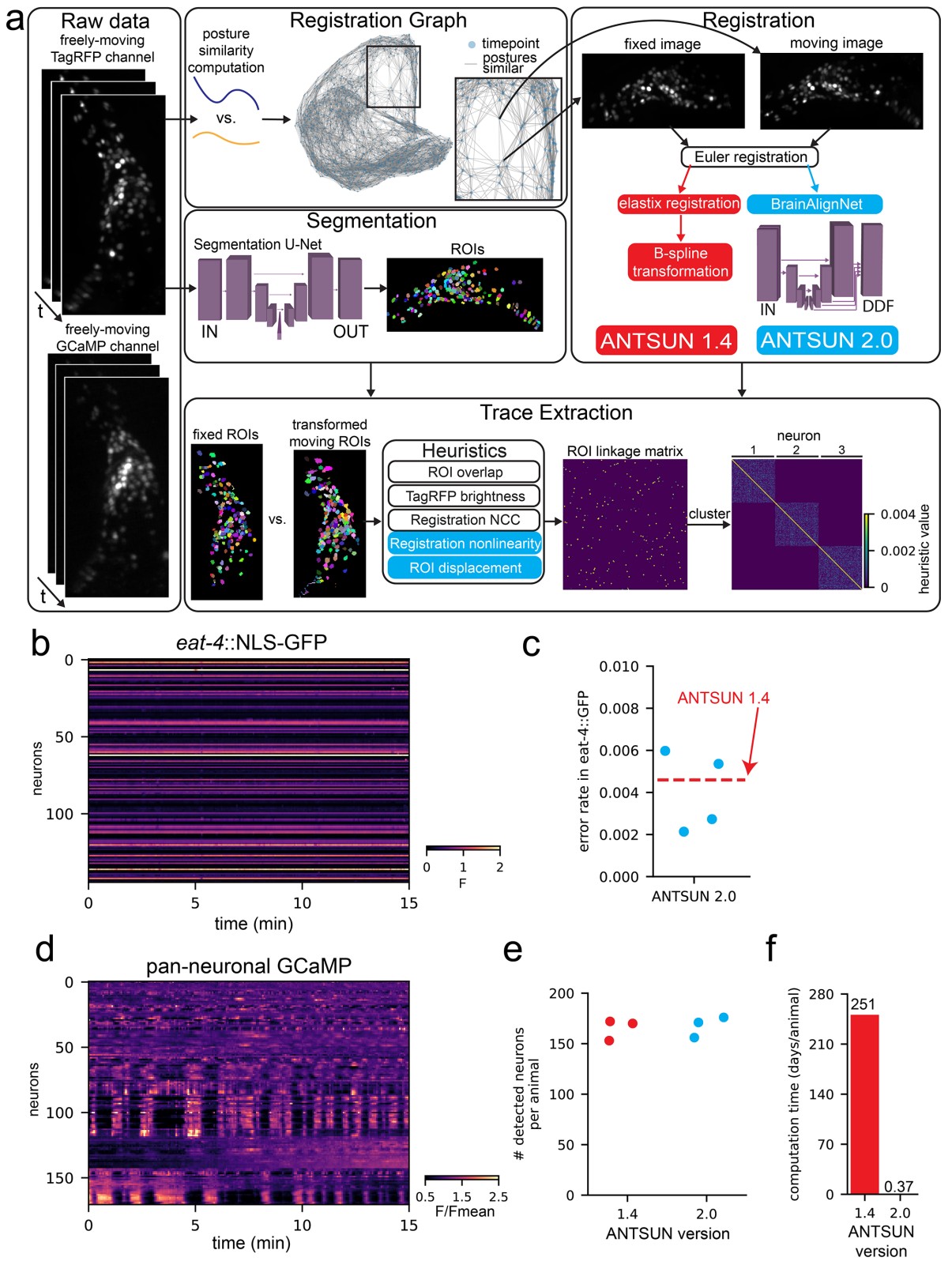

**Figure 2.** BrainAlignNet supports calcium trace extraction with high accuracy and high SNR. (**A**) Diagram of ANTSUN 1.4 and 2.0, which are two full calcium trace extraction pipelines that only differ with regards to image registration. Raw tagRFP channel data is input into the pipeline, which submits image pairs with similar worm postures for registration using either elastix (ANTSUN 1.4; red) or BrainAlignNet (ANTSUN 2.0; blue). The registration is used to transform neuron ROIs identified by a segmentation U-Net (the cuboid diagram is represented as in *Figure 1A*). These are input into a heuristic

*Figure 2 continued on next page*

*Figure 2 continued*

function (ANTSUN 2.0-specific heuristics shown in blue) which defines an ROI linkage matrix. Clustering this matrix then yields neuron identities. (**B**) Sample dataset from an *eat-4*::NLS-GFP strain, showing ratiometric (GFP/tagRFP) traces without any further normalization. This strain has some GFP +neurons (bright horizontal lines) as well as some GFP- neurons (dark horizontal lines, which have $F\sim0$). Registration artifacts between GFP+ and GFP- neurons would be visible as bright points in GFP- traces or dark points in GFP + traces. (**C**) Error rate of ANTSUN 2.0 registration across four *eat-4*::NLS-GFP animals, computed based on mismatches between GFP+ and GFP- neurons in the *eat-4*::NLS-GFP strain. The dashed red line shows the error rate of ANTSUN 1.4. Individual dots are different recorded datasets. Note that all error rates are <1%. (**D**) Sample dataset from a pan-neuronal GCaMP strain, showing F/Fmean fluorescence. Robust calcium dynamics are visible in most neurons. (**E**) Number of detected neurons across three pan-neuronal GCaMP animals for the two different ANTSUN versions (1.4 or 2.0). Individual dots are individual recorded datasets. (**F**) Computation time to process one animal based on ANTSUN version (1.4 or 2.0). ANTSUN 1.4 was run on a computing cluster that provided an average of 32 CPU cores per registration problem; computation time is the total number of CPU hours used (i.e.: the time it would have taken to run ANTSUN 1.4 registration locally on a comparable 32-core machine). ANTSUN 2.0 was run locally on NVIDIA A4000, A5500, and A6000 graphics cards.

The online version of this article includes the following figure supplement(s) for figure 2:

**Figure supplement 1.** Characterization of pan-neuronal GFP datasets processed by ANTSUN 2.0.

We also quantified other aspects of performance on GCaMP datasets (example GCaMP datasets in *Figure 2D*). ANTSUN 2.0 successfully extracted traces from a similar number of neurons as ANTSUN 1.4 (*Figure 2E*). However, while ANTSUN 1.4 requires 250 CPU days per dataset for registration, ANTSUN 2.0 only requires 9 GPU hours, reflecting a >600 fold increase in computation speed (*Figure 2F*). These results suggest that ANTSUN 2.0, which uses BrainAlignNet, provides a massive speed improvement in extracting neural data from GCaMP recordings without compromising the accuracy of the data.

## BrainAlignNet can be used to register neurons from moving jellyfish

We next sought to examine whether BrainAlignNet could be purposed to align neurons from other animals in distinct imaging paradigms. The hydrozoan jellyfish *Clytia hemisphaerica* has recently been developed as a genetically and optically tractable model for systems and evolutionary neuroscience (*Cunningham et al., 2024*). It presents a valuable test of the generalizability of our methods as the neuronal morphology, radially symmetric body plan, and centripetal motor output are all distinct from our *C. elegans* preparations above.

To test the generalizability of BrainAlignNet, we collected a set of videos in which *Clytia* were gently restrained under cover glass: neurons were restricted to a single z-plane in an epifluorescence configuration. In this setting, *Clytia* warps significantly as it moves and behaves, necessitating non-rigid motion correction. This paradigm provides an opportunity to relate neural activity to behavior, but even in this context, there are non-rigid deformations that make neuron tracking over time very challenging. Further, in this preparation, neurons have the potential to overlap in the single z-plane. This makes it important to perform full image alignment on all frames of a video, that is aligning all data to a single frame (note that this is different from the approach in *C. elegans*, see *Figure 2A*). We tested whether BrainAlignNet could facilitate this demanding case of neuron alignment.

To assess BrainAlignNet's performance, we utilized recordings of a sparsely labeled transgenic line where neuron tracking is solvable using classic point-tracking across adjacent frames. This allowed us to determine ground-truth neuron positions (i.e. centroids) that could be used to rigorously quantify alignment accuracy. Specifically, we collected videos of transgenic *Clytia* medusae expressing mCherry under control of the RFamide neuropeptide promoter, which expresses in roughly 10% of neurons distributed throughout the animal's body (*Weissbourd et al., 2021*). We formulated the alignment problem as follows. All images were Euler-aligned to a single reference frame. While Euler registration removes much of the macro displacement of the animal, it is entirely insufficient for trace extraction as it was unable to account for the myriad non-rigid deformations occurring across the jellyfish nerve ring (*Figure 3A* shows example). The network was as described above, with only slight modifications to mask out DDF deformations in the *z* axis and ignore regularization across *z* slices (since the image was 2-D). We then trained on 300 total image pairs from three animals using only image loss and regularization loss (we found that centroid alignment loss was unnecessary in this more constrained 2D search space). We then evaluated image alignment (*Figure 3B*) and centroid alignment (*Figure 3C*) on 25,997 frames from three separate animals withheld from the training data. Compared to the Euler-aligned images, BrainAlignNet led to a dramatic increase in image alignment

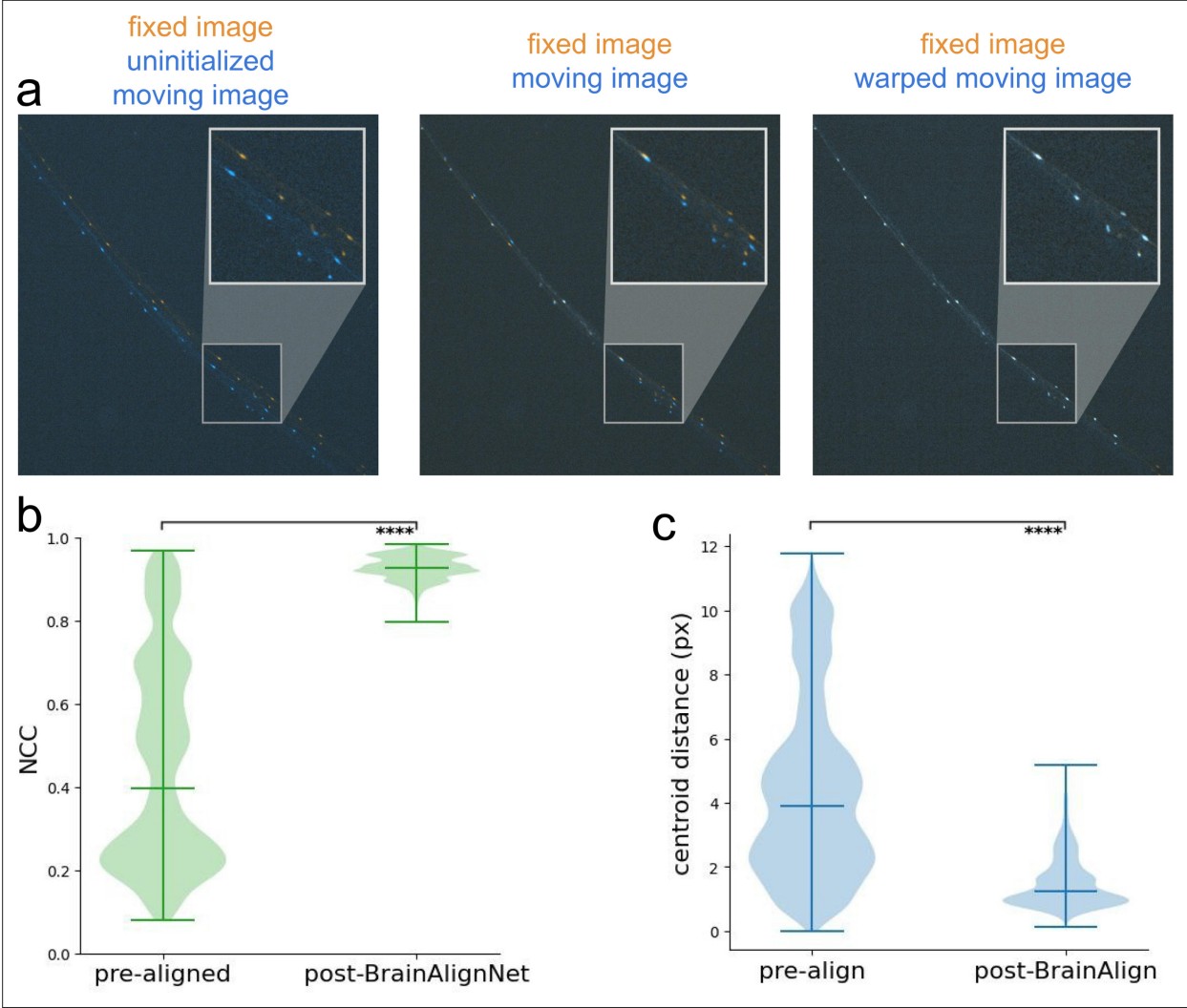

**Figure 3.** BrainAlignNet can be used to perform neuron alignment in jellyfish. (**A**) Example of image registration on a pair of mCherry images (from a testing animal, withheld from training data), composed by overlaying a moving image (blue) on the fixed image (orange). While the fixed image remains untransformed, the uninitialized moving image (left) gets warped by a Euler transformation (middle) and a BrainAlignNet-generated DDF (right) to overlap with the fixed image. (**B**) Evaluation of registration performance by examining mCherry image alignment (via NCC) on testing datasets before and after registration with BrainAlignNet. 'pre-align' shows alignment statistics on images after Euler alignment, but before neural network registration. Here, we evaluated all registration problems for all three animals in the testing set. As in *Figure 1*, NCC quantifies alignment of the fixed and warped moving RFP images, where a score of 1 indicates perfect alignment. All violin plots are accompanied by lines indicating the minimum, mean, and maximum values. ****p<0.0001, two-tailed Wilcoxon signed rank test. For both 'pre-align' and 'post-BrainAlignNet', n=25,997 registration problems (from three animals). (**C**) Evaluation of registration performance by examining neuron alignment (measured via distance between matched centroids) on testing datasets before and after BrainAlignNet registration. Centroid distance is measured as the mean Euclidean distance between the centroids of all neurons in the fixed image and the centroids of the corresponding neurons in the warped moving image; a distance of 0 indicates perfect alignment. ****p<0.0001, two-tailed Wilcoxon signed rank test. For both 'pre-align' and 'post-BrainAlignNet', n=25,997 registration problems (from three animals).

The online version of this article includes the following figure supplement(s) for figure 3:

**Figure supplement 1.** BrainAlignNet performance on additional withheld jellyfish data.

and centroid alignment, ultimately aligning matched centroids within 1.51 pixels of each other on average (*Figure 3B–C*). Additionally, we found comparable performance across the withheld frames of the training animals (*Figure 3—figure supplement 1*). These results obtained in jellyfish show that, with only light modification, a version of BrainAlignNet can produce neuron alignment in a radically different organism at a quality level matching that observed in our *C. elegans* data – aligning matched neurons within ~1.5 pixels of each other.

## AutoCellLabeler: a neural network that automatically annotates >100 neuron classes in the *C. elegans* head from multi-spectral fluorescence

We next turned our attention to annotating the identities of the recorded neurons in brain-wide calcium imaging data. *C. elegans* neurons have fairly stereotyped positions in the heads of adult animals, though fully accurate inference of neural identity from position alone has not been shown to be possible. Fluorescent reporter gene expression using well-defined genetic drivers can provide additional information. The NeuroPAL strain is especially useful in this regard. It expresses pan-neuronal NLS-tagRFP, but also has expression of NLS-mTagBFP2, NLS-CyOFP1, and NLS-mNeptune2.5 under a set of well-chosen genetic drivers (example image in *Figure 4A Yemini et al., 2021*). With proper training, humans can manually label the identities of most neurons in this strain using neuron position and multi-spectral fluorescence. For most of the brain-wide recordings collected using our calcium imaging platform, we used a previously characterized strain with a pan-neuronal NLS-GCaMP7F transgene crossed into NeuroPAL (*Atanas et al., 2023*). While freely moving recordings were conducted with only NLS-GCaMP and NLS-tagRFP data acquisition, animals were immobilized at the end of each recording to capture multi-spectral fluorescence. Humans could manually label many neurons' identities in these multi-spectral images, and the image registration approaches described above could map the ROIs in the immobilized data to ROIs in the freely moving recordings in order to match neuron identity to GCaMP traces.

Manual annotation of NeuroPAL images is time-consuming. First, to perform accurate labeling, the individual needs substantial training. Even after being trained, labeling all ROIs in one NeuroPAL animal can take 3–5 hr. In addition, different individuals have different degrees of knowledge or confidence in labeling certain cell classes. For these reasons, it was desirable to automate NeuroPAL labeling using datasets that had previously been labeled by a panel of human labelers. In particular, the labels that they provided with a high degree of confidence would be most useful for training an automated labeling network. Previous studies have developed statistical approaches for semi-automated labeling to label neural identity from NeuroPAL images, but the maximum precision that we are aware of is 90% without manual correction (*Yemini et al., 2021*).

We trained a 3-D U-Net (*Wolny et al., 2020*) to label the *C. elegans* neuron classes in a given NeuroPAL 3-D image. As input, the network received four fluorescent 3-D images from the head of each worm: pan-neuronal NLS-tagRFP, plus the NLS-mTagBFP2, NLS-CyOFP1, and NLS-mNeptune2.5 images that label stereotyped subsets of neurons (*Figure 4A*). During training, the network also received the human-annotated labels of which pixels belong to which neurons. Humans provided ROI-level labels and the boundaries of each ROI were determined using a previously described neuron segmentation network (*Atanas et al., 2023*) trained to label all neurons in a given image (agnostic to their identity). Finally, during training, the network also received an array indicating the relative weight to assign each pixel during training (*Figure 4B*). This was incorporated into a pixel-weighted cross-entropy loss function (lower values indicate more accurate labeling of each pixel), summing across the pixels in a weighted manner. Pixel weighting was adjusted as follows: (1) background was given extremely low weight; (2) ROIs that humans were not able to label were given extremely low weight; (3) all other ROIs received higher weight proportional to the subjective confidence that the human had in assigning the label to the ROI and the rarity of the label (see Materials and methods for exact details). Regarding this latter point, neurons that were less frequently labeled by human annotation received higher weight so that the network could potentially learn how to classify these neurons from fewer labeled examples.

We trained the network over 300 epochs using a training set of 81 annotated images and a validation set of 10 images (*Figure 4C*). Because the size of the training set was fairly small, we augmented the training data using both standard image augmentations (rotation, flipping, adding Gaussian noise, etc.) and a custom augmentation where the images were warped in a manner to approximate worm head bending (see Materials and methods). Overall, the goal was for this network to be able to annotate neural identities in worms in any posture provided that they were roughly oriented length-wise in a 284 x 120 x 64 (x, y, z) image. Because this *Auto*matic *Cell Label*ing Network (**AutoCellLabeler**) labels individual pixels, it was necessary to convert these pixel-wise classifications into ROI-level classifications. AutoCellLabeler outputs its confidence in its label for each pixel, and we noted that the network's confidence for a given ROI was highest near the center of the ROI (*Figure 4D*). Therefore, to determine ROI-level labels, we took a weighted average of the pixel-wise labels within an ROI,

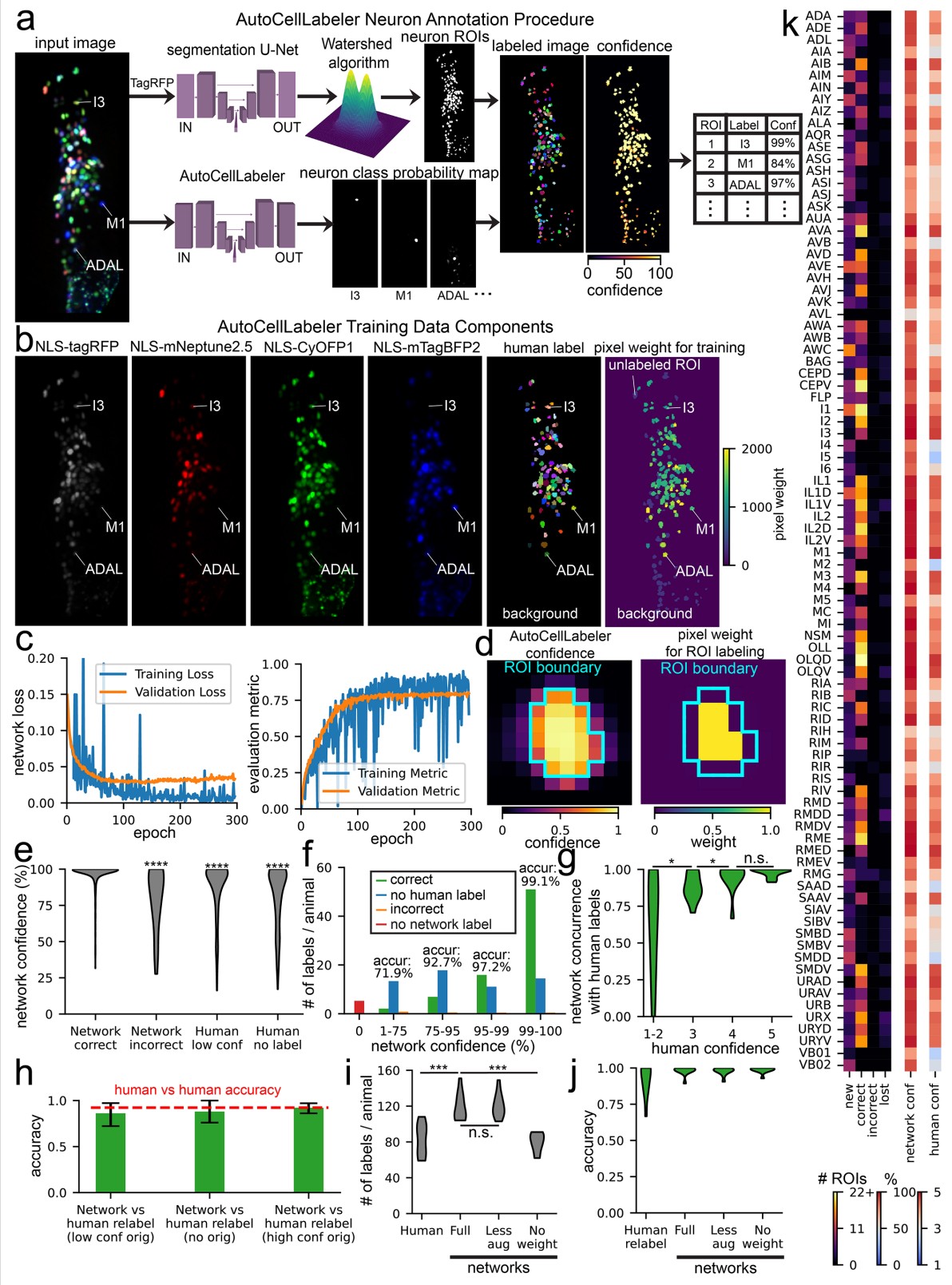

**Figure 4.** The AutoCellLabeler Network can automatically annotate >100 neuronal cell types in the *C. elegans* head. (**A**) Procedure by which AutoCellLabeler generates labels for neurons. First, the tagRFP component of a multi-spectral image is passed into a segmentation neural network, which extracts neuron ROIs, labeling each pixel as an arbitrary number with one number per neuron. Then, the full multi-spectral image is input into AutoCellLabeler, which outputs a probability map. This probability map is applied to the ROIs to generate labels and confidence values for those

*Figure 4 continued on next page*

*Figure 4 continued*

labels. The network cuboid diagrams are represented as in *Figure 1A*. (**B**) AutoCellLabeler's training data consists of a set of multi-spectral images (NLS-tagRFP, NLS-mNeptune2.5, NLS-CyOFP1, and NLS-mTagBFP2), human neuron labels, and a pixel weighting matrix based on confidence and frequency of the human labels that controls how much each pixel is weighted in AutoCellLabeler's loss function. (**C**) Pixel-weighted cross-entropy loss and pixel-weighted IoU metric scores for training and validation data. Cross-entropy loss captures the discrepancy between predicted and actual class probabilities for each pixel. The IoU metric describes how accurately the predicted labels overlap with the ground truth labels. (**D**) During the label extraction procedure, AutoCellLabeler is less confident of its label on pixels near the edge of ROI boundaries. Therefore, we allow the central pixels to have much higher weight when determining the overall ROI label from pixel-level network output. (**E**) Distributions of AutoCellLabeler's confidence across test datasets based on the relationship of its label to the human label ('Correct'=agree, 'Incorrect'=disagree, 'Human low conf'=human had low confidence, 'Human no label' – human did not even guess a label for the neuron). ****p<0.0001, as determined by a Mann-Whitney U Test between the indicated condition and the 'Correct' condition where the network agreed with the human label; n=835, 25, 322, 302 labels (from 11 animals) for the conditions 'Correct', 'Incorrect', 'Human low conf', 'Human no label', respectively; U=16,700, 202,691, 210,797 for 'Incorrect', 'Human low conf', 'Human no label' vs 'Correct', respectively. (**F**) Categorization of neurons in test datasets based on AutoCellLabeler's confidence. Here 'Correct' and 'Incorrect' are as in (**E**), but 'No human label' also includes low-confidence human labels. Printed percentage values are the accuracy of AutoCellLabeler on the corresponding category, computed as $\frac{correct}{correct+incorrect}$ (**G**) Distributions of accuracy of AutoCellLabeler's high confidence (>75%) labels on neurons across test datasets based on the confidence of the human labels. n.s. not significant, *p<0.05, as determined by a paired permutation test comparing mean differences (n=11 test datasets). (**H**) Accuracy of AutoCellLabeler compared with high-confidence labels from new human labelers on neurons in test datasets that were labeled at low confidence, not at all, or at high confidence by the original human labelers. Error bars are bootstrapped 95% confidence intervals. A dashed red line shows accuracy of new human labelers relative to the old human labelers, when both gave high confidence to their labels. There was no significant difference between the human vs human accuracy and the network accuracy for any of these categories of labels, determined via two-tailed empirical p-values from the bootstrapped distributions. (**I**) Distributions of number of high-confidence labels per animal over test datasets. High confidence was 4–5 for human labels and >75% for network labels. We note that we standardized the manner in which split ROIs were handled for human- and network-labeled data so that the number of detected neurons could be properly compared between these two groups. n.s. not significant, ***p<0.001, as determined by a paired permutation test comparing mean differences (n=11 animals). (**J**) Distributions of accuracy of high-confidence labels per animal over test datasets, relative to the original human labels. A paired permutation test comparing mean differences to the full network's label accuracy did not find any significance. (**K**) Number of ROIs per neuron class labeled at high confidence in test datasets that fall into each category, along with average confidence for all labels for each neuron class in those test datasets. 'New' represents ROIs that were labeled by the network as the neuron and were not labeled by the human. 'Correct' represents ROIs that were labeled by both AutoCellLabeler and the human as that neuron. 'Incorrect' represents ROIs that were labeled by the network as that neuron and were labeled by the human as something else. 'Lost' represents ROIs that were labeled by the human as that neuron and were not labeled by the network. 'Network conf' represents the average confidence of the network for all its labels of that neuron. 'Human conf' represents the average confidence of the human labelers for all their labels of that neuron. Neuron classes with high values in the 'Correct' column and low values in the 'Incorrect' column indicate a very high degree of accuracy in AutoCellLabeler's labels for those classes. If those classes also have a high value in the 'New' column, it could indicate that AutoCellLabeler is able to find the neuron with high accuracy in animals where humans were unable to label it.

The online version of this article includes the following figure supplement(s) for figure 4:

**Figure supplement 1.** Further characterization of the AutoCellLabeler network.

weighing the center pixels more strongly. The overall confidence of these pixel scores was also used to compute an ROI-level confidence score, reflecting the network's confidence that it labeled the ROI correctly. Finally, after all ROIs were assigned a label, heuristics were applied to identify and delete problematic labels. Labels were deleted if (1) the network already labeled another ROI as that label with higher confidence; (2) the label was present too infrequently in the network's training data; (3) the network labeled that ROI as something other than a neuron (e.g. a gut granule or glial cell, which we supplied as valid labels during training and had labeled in training data); or (4) the network confidently predicted different parts of the ROI as different labels (see Materials and methods for details).

We evaluated the performance of the network on 11 separate datasets that were reserved for testing. We assessed the accuracy of AutoCellLabeler on the subset of ROIs with high-confidence human labels (subjective confidence scores of 4 or 5, on a scale from 1 to 5). On these neurons, average network confidence was 96.8% and its accuracy was 97.1%. We furthermore observed that the network was more confident in its correct labels (average confidence 97.3%) than its incorrect labels (average confidence 80.7%; *Figure 4E*). More generally, AutoCellLabeler confidence was highly correlated with its accuracy (*Figure 4F* shows a breakdown of cell labeling at different confidences, with an inset indicating accuracy). Indeed, excluding the neurons where the network assigns low (<75%) confidence increased its accuracy to 98.1% (*Figure 4—figure supplement 1A* displays the full accuracy-recall tradeoff curve). Under this confidence threshold cutoff, AutoCellLabeler still assigned a label to 90.6% of all the ROIs that had high-confidence human labels, so we chose to delete the

low-confidence (<75%) labels altogether (see *Figure 4—figure supplement 1A* for rationale for the 75% cutoff value).

We also examined model performance on data where humans had either low confidence or did not assign a neuron label. In these cases, it was harder to estimate the ground truth. Overall, model confidence was much lower for neurons that humans labeled with low confidence (87.3%) or did not assign a label (81.3%). The concurrence of AutoCellLabeler relative to low-confidence human labels was also lower (84.1%; we note that this is not really a measure of accuracy since these 'ground-truth' labels had low confidence). Indeed, overall the network's concurrence versus human labels scaled with the confidence of the human label (*Figure 4G*).

We carefully examined the subset of ROIs where the network had high confidence (>75%), but humans had either low confidence or entered no label at all. This was quite a large set of ROIs: AutoCellLabeler identified significantly more high-confidence neurons (119/animal) than the original human labelers (83/animal), and this could conceivably reflect a highly accurate pool of network-generated labels exceeding human performance. To determine whether this was the case, we obtained new human labels by different human labelers for a random subset of these neurons. Whereas some human labels remained low-confidence, others were now labeled with high confidence (20.9% of this group of ROIs). The new human labelers also labeled neurons that were originally labeled with high confidence so that we could compare the network's performance on relabeled data where the original data was unlabeled, low confidence, or high confidence. AutoCell-Labeler's performance on all three groups was similar (88%, 86.1%, and 92.1%, respectively), which was comparable to the accuracy of humans relabeling data relative to the original high-confidence labels (92.3%; *Figure 4H*). The slightly lower accuracy on these re-labeled data is likely due to the human labeling of the original training, validation, and testing data being highly vetted and thoroughly double-checked, whereas the re-labeling that we performed just for this analysis was done in a single pass. Overall, these analyses indicate that the high-confidence network labels (119/animal) have similar accuracy regardless of whether the original data had been labeled by humans as unlabelable, low confidence, or high confidence. We note that this explains a subset of the cases where human low-confidence labels were not in agreement with network labels (*Figure 4G*). Taken together, these observations indicate that AutoCellLabeler can confidently label more neurons per dataset than individual human labelers.

We also split out model performance by cell type. This largely revealed similar trends. Model labeling accuracy and confidence were variable among the neuron types, with highest accuracy and confidence for the cell types where there were higher confidence human labels and a higher frequency of human labels (*Figure 4K*). For the labels where there were high confidence network and human labels, we generated a confusion matrix to see if AutoCellLabeler's mistakes had recurring trends (*Figure 4—figure supplement 1B*). While mistakes of this type were very rare, we observed that the ones that occurred could mostly be categorized as either mislabeling a gut granule as the neuron RMG, or mislabeling the dorsal/ventral categorization of the neurons IL1 and IL2 (e.g. mislabeling IL2D as IL2). Together, these categories accounted for 50% of all AutoCellLabeler's mistakes. We also observed that across cell types, AutoCellLabeler's confidence was highly correlated with human confidence (*Figure 4—figure supplement 1C*), suggesting that the main limitations of model accuracy are due to limited human labeling accuracy and confidence.

To provide better insights into which network features were critical for its performance, we trained additional networks lacking some of AutoCellLabeler's key features. To evaluate these networks, we considered both the number of high-confidence labels assigned by AutoCellLabeler and the accuracy of those labels measured against high-confidence human labels. Surprisingly, a network that was trained with only standard image augmentations (i.e. lacking the custom augmentation to bend the images in a manner that approximates a worm head bend) had similar performance (*Figure 4I*). However, a network that was trained without a pixel-weighting scheme (i.e. where all pixels were weighted equally) provided far fewer high-confidence labels. This suggests that devising strategies for pixel weighting is critical for model performance, though our custom augmentation was not important. Interestingly, all trained networks had similar accuracy (*Figure 4J*) on their high-confidence labels, suggesting that the network architecture in all cases is able to accurately assess its confidence.

## Automated annotation of *C. elegans* neurons from fewer fluorescent labels and in different strains

We examined whether the full group of fluorophores was critical for AutoCellLabeler performance. This is a relevant question because (i) it is laborious to make, inject, and annotate a large number of plasmids driving fluorophore expression and (ii) the large number of plasmids in the NeuroPAL strain has been noted to adversely impact the animals' growth and behavior (*Atanas et al., 2023*; *Yemini et al., 2021*; *Sharma et al., 2024*). To test whether fewer fluorescent labels could still facilitate automatic labeling, we trained four additional networks: one that only received the pan-neuronal tagRFP image as input, and three that received pan-neuronal tagRFP plus a single other fluorescent channel (CyOFP, tag-mBFP2, or mNeptune). As we still had the ground-truth labels based on humans viewing the full set of fluorophores, the supervised labels were identical to those supplied to the full network.

We evaluated the performance of these models by quantifying the number of high-confidence labels that each network provided in each testing dataset (*Figure 5A*) and the accuracy of these labels measured against high-confidence human labels (*Figure 5B*). We found that all four networks had attenuated performance relative to the full AutoCellLabeler network, which was almost entirely explainable by these networks having lower confidence in their labels, since network accuracy was always consistent with its confidence (*Figure 5—figure supplement 1A*). Of the four attenuated networks, the tagRFP +CyOFP network performance (107 neurons per animal labeled at 97.4% accuracy) was closest to the full network in its performance. Given that there are >20 mTagBFP2 and mNeptune plasmids in the full NeuroPAL strain, these results raise the possibility that a smaller set of carefully chosen plasmids could permit training of a network with equal performance to the full network that we trained here.

We did not expect the tagRFP-only network to perform well, since the task of labeling tagRFP-only images is nearly impossible for humans. Surprisingly, this network still exhibited relatively high performance, with an average of 94 high-confidence neurons per animal and 94.8% accuracy on those neurons. On most neuron classes, it behaved nearly as well as the full network, though there are 10–20 neuron classes that it is much worse at labeling, such as ASG, IL1, and RMG (*Figure 5C*). This suggests a high level of stereotypy in neuron position, shape, and/or RFP brightness for many neuron types. Since this network only requires the red channel fluorescence, it could be used directly on freely moving data, which has only GCaMP and tagRFP channel data. Since the tagRFP-only network was trained only on high-SNR images collected from immobilized animals, we first checked that the network was able to generalize outside its training distribution to single images with lower SNR (example images in *Figure 5—figure supplement 1B*). It was able to label 79 high-confidence neurons per animal at 95.2% accuracy on the lower SNR images (*Figure 5A–B*, right). We then investigated whether allowing the network to access different postures of the same animal improved its accuracy. Specifically, we evaluated the tagRFP-only network on 100 randomly selected timepoints in the freely moving data of each animal (example images in *Figure 5—figure supplement 1B*). We then related these 100 network labels to the human labels, which could be easily determined, since ANTSUN registers freemoving images back to the immobilized NeuroPAL images that had been labeled by humans. We averaged the 100 network labels to obtain the most likely network label for each neuron, as well as the average confidence for that label. To properly compare network versions, we determined how many neurons could be labeled at any given target labeling accuracy – for example, how many neurons the network can label and still achieve 95% accuracy (*Figure 5D*; changing the threshold network confidence value to include a given label allowed us to determine these full curves). This analysis revealed that averaging network labels across the 100 timepoints improved network performance, though only modestly. These results suggest that single-color labels can be used to train networks to a high level of performance, but additional fluorescence channels further improve performance.

The strong performance of the tagRFP-only network on out-of-domain lower SNR images suggests an impressive ability of the AutoCellLabeler network to generalize across different modalities of data. This raised the possibility that it may be possible to use this network architecture to build a foundation model of *C. elegans* neuron annotation that works across strains and imaging conditions. As a first step to explore this idea, we investigated to what extent the tagRFP-only network could generalize to other strains of *C. elegans* besides the NeuroPAL strain. We used our previously described SWF415 strain, which contains pan-neuronal NLS-GCaMP7F, pan-neuronal NLS-mNeptune2.5, and sparse tagRFP expression (*Atanas et al., 2023*). Notably, the pan-neuronal promoter used in this

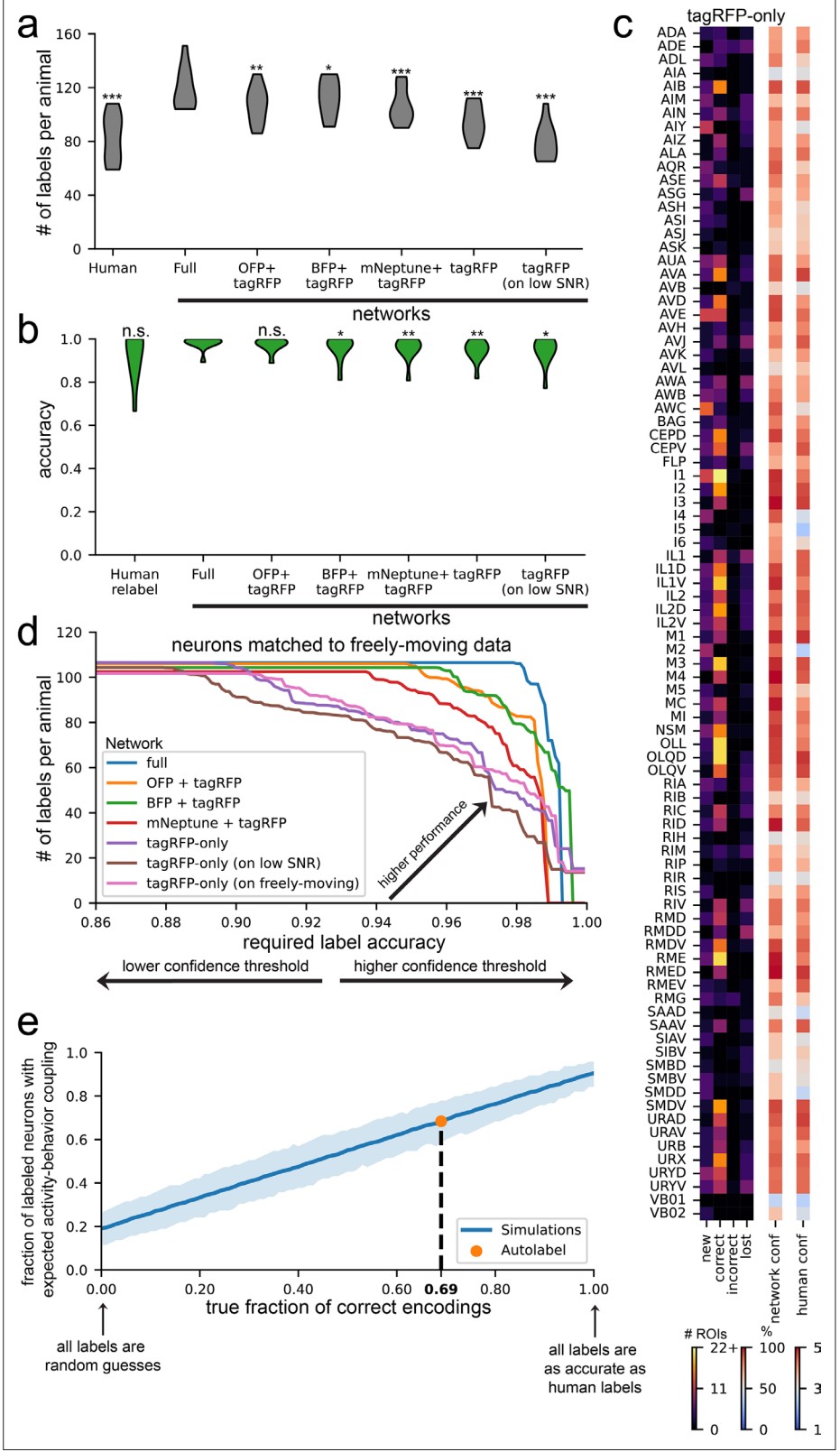

**Figure 5.** Variants of AutoCellLabeler can annotate neurons from fewer fluorescent channels and in different strains. (**A**) Distributions of number of high-confidence labels per animal over test datasets for the networks trained on the indicated set of fluorophores. The 'tagRFP (on low SNR)' column corresponds to a network that was trained on high-SNR, tagRFP-only data and tested on low-SNR tagRFP data due to shorter exposure times

*Figure 5 continued on next page*

*Figure 5 continued*

in freely moving animals. *p<0.05, **p<0.01, ***p<0.001, as determined by a paired permutation test comparing mean differences to the full network (n=11 animals). (**B**) Distributions of accuracy of high-confidence labels per animal over test datasets for the networks trained on the indicated set of fluorophores. The 'tagRFP (on low SNR)' column is as in (**A**). n.s. not significant, *p<0.05, **p<0.01, as determined by a paired permutation test comparing mean differences to the full network (n=11 animals). (**C**) Same as *Figure 4K*, except for the tagRFP-only network. (**D**) Accuracy vs detection tradeoff for various AutoCellLabeler versions. For each network, we can set a confidence threshold above which we accept labels. By varying this threshold, we can produce a tradeoff between accuracy of accepted labels (*x*-axis) and number of labels per animal (*y*-axis) on test data. Each curve in this plot was generated in this manner. The 'tagRFP-only (on low SNR)' values are as in (**A**). The 'tagRFP-only (on freely moving)' values come from evaluating the tagRFP-only network on 100 randomly-chosen timepoints in the freely moving (tagRFP) data for each test dataset. The final labels were then computed on each immobilized ROI by averaging together the 100 labels and finding the most likely label. To ensure fair comparison to other networks, only immobilized ROIs that were matched to the freely moving data were considered for any of the networks in this plot. (**E**) Evaluating the performance of tagRFP-only AutoCellLabeler on data from another strain SWF415, where there is pan-neuronal NLS-GCaMP7f and pan-neuronal NLS-mNeptune2.5. Notably, the pan-neuronal promoter used for NLS-mNeptune2.5 differs from the pan-neuronal promoter used for NLS-tagRFP in NeuroPAL. Performance here was quantified by computing the fraction of network labels with the correct expected activity-behavior relationships in the neuron class (y-axis; quantified by whether an encoding model showed significant encoding; see Materials and methods). For example, when the label was the reverse-active AVA neuron, did the corresponding calcium trace show higher activity during reverse? The blue line shows the expected fraction as a function of the true accuracy of the network (x-axis), computed via simulations (see Materials and methods). The orange circle shows the actual fraction when AutoCellLabeler was evaluated on SWF415. Based on this, the dashed line shows estimated true accuracy of this labeling.

The online version of this article includes the following figure supplement(s) for figure 5:

**Figure supplement 1.** Further characterization of the different AutoCellLabeler variants.

strain for NLS-mNeptune expression (P*rimb-1*) is distinct from the pan-neuronal promoter that drives NLS-tagRFP expression in NeuroPAL (a synthetic promoter), so the levels of red fluorescence are likely to be different across these strains. Since humans do not know how to label neurons in SWF415 (it lacks the NeuroPAL transgene), we did a more limited analysis by analyzing network labels for a subset of neurons that have highly reliable activity dynamics with respect to behavior (AVA, AVE, RIM, and AIB encode reverse locomotion; RIB, AVB, RID, and RME encode forward locomotion; SMDD encodes dorsal head curvature; and SMDV and RIV encode ventral head curvature; *Atanas et al., 2023*; *Kato et al., 2015*; *Li et al., 2014*; *Chalfie et al., 1985*; *Gordus et al., 2015*; *Luo et al., 2014*; *Wang et al., 2020*; *Ben Arous et al., 2010*; *Lim et al., 2016*; *Hallinen et al., 2021*; *Ray and Gordus, 2025*). Specifically, we asked whether neurons labeled in SWF415 recordings with high confidence by the network had the behavior encoding properties typical of the neuron, assessed via analysis of the GCaMP traces from that neuron. Our previously described CePNEM model (*Atanas et al., 2023*) was used to determine whether each labeled neuron encoded forward/reverse locomotion or dorsal/ventral head curvature. The network provided high-confidence labels for an average of 7.4/21 of these neurons per animal, and the encoding properties of these neurons matched expectations 68% of the time (randomly labeled neurons had a match of 19%). However, it was possible for the network to (i) incorrectly label a neuron as another neuron that happened to have the same encoding; or (ii) correctly label a neuron that CePNEM lacked statistical power to declare an encoding for. We accounted for these effects via simulations (see Materials and methods), which estimated that the actual labeling accuracy of the network on SWF415 was 69% (*Figure 5E*). This is substantially lower than this network's accuracy on similar images from the NeuroPAL strain (i.e. the strain used to train the network), where an average of 12.5 of these neurons were labeled per animal with 97.1% accuracy. Nevertheless, this analysis indicates that AutoCellLabeler has a reasonable ability to generalize to strains with different genetic drivers and fluorophores, suggesting that in the future it may be worthwhile to pursue building a foundation model that labels *C. elegans* neurons across many strains.

# A neural network (CellDiscoveryNet) that facilitates unsupervised discovery of >100 cell types by aligning data across animals

Annotation of cell types via supervised learning (using human-provided labels) is fundamentally limited by prior knowledge and human throughput in labeling multi-spectral imaging data. In principle, unsupervised approaches that can automatically identify stereotyped cell types would be preferable. Thus, we next sought to train a neural network to perform unsupervised discovery of the cell types of the *C. elegans* nervous system (*Figure 6A*). If successful, these approaches could be useful for labeling of mutant genotypes, new permutations of NeuroPAL, or even related species, where it might be laborious to painstakingly characterize new multispectral lines and generate >100 labeled datasets. In addition, such an approach would be useful in more complex animals that do not yet have complete catalogs of cell types.

To facilitate unsupervised cell type discovery, we trained a network to register different animals' multi-spectral NeuroPAL imaging data to one another. Successful alignment of cells across all recorded animals would amount to unsupervised cell type annotation, since the cells that align across animals would comprise a functional 'cluster' corresponding to the same cell type identified in different animals (we note though that each cell type, or cluster, here would still need to be assigned a name). The architecture of this network was similar to BrainAlignNet, but the training data here consisted of pairs of four-color NeuroPAL images from two different animals and the network was tasked with aligning all four fluorescent channels (*Figure 6B*). No cell type positions (i.e. centroids) or neuronal identities were provided to the network during training. Regularization and augmentation were similar to that of BrainAlignNet (see Materials and methods). Training and validation data were comprised of 91 animals' datasets, which gave rise to 3285 unique pairs for alignment; 11 animals were withheld for testing (the same test set as for AutoCellLabeler). The network was trained over 600 epochs (*Figure 6C*). In the analyses below, we characterize performance on training data and withheld testing data, describing any differences. We note that, in contrast to the networks described above, high performance on training data is still useful in this case, since the only criterion for success in unsupervised learning is successful alignment (i.e. even if all data need to be used for training to do so). Strong performance on testing data is still more desirable though, since it is less efficient to train different networks over and over as new data are incorporated into the full dataset.

We first characterized the ability of this Unsupervised **Cell Discovery Net**work (**CellDiscoveryNet**) to align images of different animals. Image alignment was reasonably high for all four fluorescent NeuroPAL channels with a median NCC of 0.80 overall (*Figure 6D*). Alignment accuracy was nearly equivalent in training and testing data (*Figure 6D*). We also examined how well the centroid positions of defined cell types were aligned, utilizing our prior knowledge of neurons' locations from human labels (*Figure 6E*). We computed this metric only on cell types that were identified with high confidence in both of the images of a given registration problem. The median centroid distance was 7.2 pixels, with similar performance on training and testing data. This was initially rather disappointing, as it suggested that the majority of neurons were not being placed at their correct locations during registration. However, we observed two important properties of the centroid alignments. First, the distribution of centroid distances was bimodal (*Figure 6E*) – the 20th percentile centroid distance was only 1.4 pixels, which corresponds to a correct neuron alignment. Second, the median centroid distance decreased to 3.3 for registration problems with high (> 90th percentile = 0.85) NCC scores on the images. Together, these observations suggest that CellDiscoveryNet correctly aligns neurons some of the time.

We next sought to differentiate the neuron alignments where CellDiscoveryNet was correct from those where it was incorrect. To accomplish this, we adapted our ANTSUN pipeline (described in *Figure 2*) to use CellDiscoveryNet instead of BrainAlignNet. This modified ANTSUN 2 U (**U**nsupervised) takes as input multi-spectral data from many animals instead of monochrome images from different time points of the same animal. This approach then allows us to effectively cluster neurons that might be the same neuron found across different animals. We ran CellDiscoveryNet on pairs of images and used the resulting DDFs to align the corresponding segmented neuron ROIs. We then constructed an N-by-N matrix (N is the number of all segmented neuron ROIs in all animals, i.e. approximately # ROIs * # animals). Entries in the matrix are zero if the two neurons were in an image pair that was never registered or if the two neurons did not overlap at all in the registered image pair. Otherwise, a heuristic value indicating the likelihood that the neurons are the same was entered into

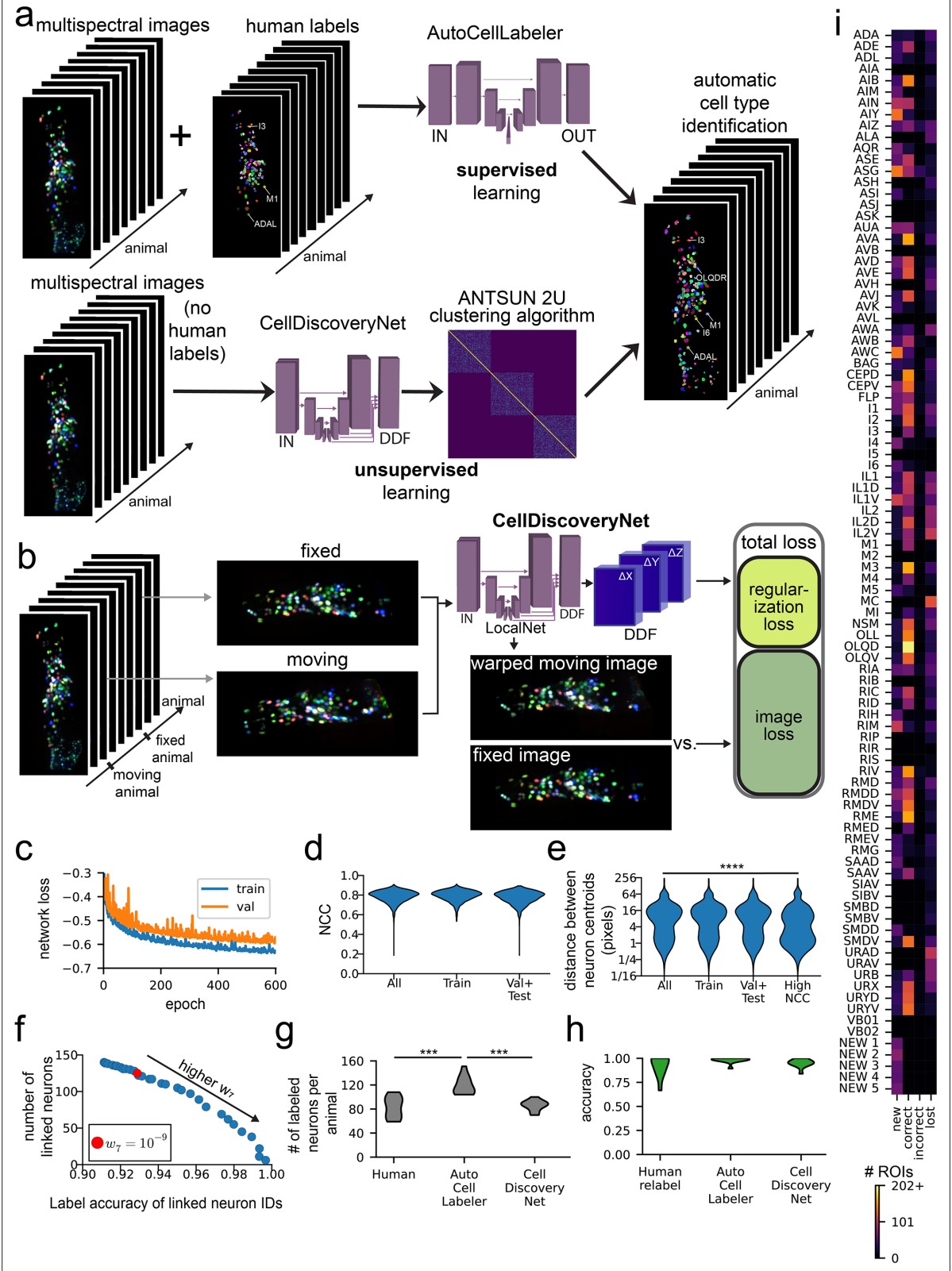

**Figure 6.** CellDiscoveryNet and ANTSUN 2 U can perform unsupervised cell type discovery by analyzing data across different *C. elegans* animals. (**A**) A schematic comparing the approaches of AutoCellLabeler and CellDiscoveryNet. AutoCellLabeler uses supervised learning, taking as input both images and manual labels for those images, and learns to label neurons accordingly. CellDiscoveryNet uses unsupervised learning and can learn to label neurons after being trained only on images (with no labels provided). (**B**) CellDiscoveryNet training pipeline. The network takes as input two

*Figure 6 continued on next page*

*Figure 6 continued*

multi-spectral NeuroPAL images from two different animals. It then outputs a Dense Displacement Field (DDF), which is a coordinate transformation between the two images. It warps the moving image under this DDF, producing a warped moving image that should ideally look very similar to the fixed image. The dissimilarity between these images is the image loss component of the loss function, which is added to the regularization loss that penalizes non-linear image deformations present in the DDF. (**C**) Network loss curves. Both training and validation loss curves start to plateau around 600 epochs. (**D**) Distributions of normalized cross-correlation (NCC) scores comparing the CellDiscoveryNet predictions (warped moving images) and the fixed images for each pair of registered images. These NCCs were computed on all four channels simultaneously, treating the entire image as a single 4D matrix for this purpose. The 'Train' distribution contains the NCC scores for all pairs of images present in CellDiscoveryNet's training data, while the 'Val +Test' distribution contains any pair of images that was not present in its training data. (**E**) Distributions of centroid distance scores based on human labels. These are computed over all (moving, fixed) image pairs on all neurons with high-confidence human labels in both moving and fixed images. The centroid distance scores represent the Euclidean distance from the network's prediction for where the neuron was and its correct location as labeled by the human. Values of a few pixels or less likely roughly indicate that the neuron was mapped to its correct location, while large values mean the neuron was mis-registered. The 'Train' and 'Val +Test' distributions are as in (**D**). The 'High NCC' distribution is from only (moving, fixed) image pairs where the NCC score was greater than the 90th percentile of all such NCC scores. ****p<0.0001, Mann-Whitney U-Test comparing All versus High NCC (n=5048 vs 486 image pairs, $U = 1.678 \times 10^6$). (**F**) Labeling accuracy vs number of linked neurons tradeoff curve. Accuracy is the fraction of linked ROIs with labels matching their cluster's most frequent label (see Materials and methods). The number of linked neurons is the total number of distinct clusters; each cluster must contain an ROI in more than half of the animals to be considered a cluster. The parameter $w_7$ describes when to terminate the clustering algorithm – higher values mean the clustering algorithm terminates earlier, resulting in more accurate but fewer detections. Red dot is the selected value $w_7 = 10^{-9}$ where 125 clusters were detected with 93% labeling accuracy. (**G**) Number of neurons labeled per animal in the 11 testing datasets. This plot compares the number of neurons labeled as follows: human labels with 4–5 confidence, AutoCellLabeler labels with 75% or greater confidence, and CellDiscoveryNet with ANTSUN 2 U labels with parameter $w_7 = 10^{-9}$. ***p<0.001, as determined by a paired permutation test comparing mean differences (n=11 animals). (**H**) Accuracy of neuron labels in the 11 testing datasets. This plot defines the original human confidence 4–5 labels as ground truth. 'Human relabel' refers to confidence 4–5 labels done by different humans (independently from the first set of human labels). AutoCellLabeler labels are confidence 75% or greater labels. CellDiscoveryNet labels were created by running ANTSUN 2 U with $w_7 = 10^{-9}$, and defining the correct label for each cluster to be its most frequent label. A paired permutation test comparing mean differences to the full network's label accuracy did not find any significance. (**I**) Same as *Figure 5K*, except using labels from CellDiscoveryNet with ANTSUN 2 U. The neurons 'NEW 1' through 'NEW 5' are clusters that were not labeled frequently enough by humans to be able to determine which neuron class they corresponded to, as described in the main text.

The online version of this article includes the following figure supplement(s) for figure 6:

**Figure supplement 1.** Further characterization of CellDiscoveryNet and ANTSUN 2 U performance.

---

the matrix. This heuristic included the same information as in ANTSUN 2.0 (described above), such as registration quality and ROI position similarity. The only difference was that the heuristic for tagRFP brightness similarity was replaced with a heuristic for four-channel color similarity (see Methods). Custom hierarchical clustering of the rows of this matrix then identified groups of ROIs hypothesized to be the same cell type identified in different animals.

To determine the performance of this unsupervised cell type discovery approach, we quantified both the number of cell types that were discovered (i.e. number of clusters) and the accuracy of cell type labeling within each cluster. Here, accuracy was computed by first determining the most frequent neuron label for each cell type, based on the human labels. We then determined the number of correct versus incorrect detections of this cell type for all cells that fell within the cluster. A correct detection was defined to be when the human label for that cell matched the most frequent label for that cell's cluster. The number of cell types identified and the labeling accuracy are directly related: more permissive clustering identifies more cell types, but at the cost of lower accuracy. A full curve revealing this tradeoff is shown in *Figure 6F* (see Materials and methods). Based on this curve, we selected the clustering parameters that identified 125 cell types with 93% labeling accuracy. Not every cell type is detected in every animal. On the testing data, the CellDiscoveryNet-powered ANTSUN 2 U roughly matched human-level performance in terms of accuracy and number of neurons labeled per animal (*Figure 6G–H*). However, it fell slightly short of AutoCellLabeler (*Figure 6G–H*). Overall, this analysis reveals that CellDiscoveryNet facilitates unsupervised cell type discovery with a high level of performance, matching trained human labelers. Specifically, this network, combined with our clustering approach, can identify 125 canonical cell types across animals and assign cells to these cell type classes with 93% accuracy.

We examined whether the accuracy of cell identification was different across cell types or across animals. *Figure 6I* shows the accuracy of labeling for each cell type (see *Figure 6—figure supplement 1A* for per-animal accuracy). Indeed, results were mixed: some cell types had highly accurate detections across animals (eg: OLQD and RME), whereas a smaller subset of cell types was detected

with lower accuracy (eg: AIZ and ASG), and yet other cell types were harder to assess accuracy due to a smaller number of human labels (e.g.: AIM and I4). In addition, there were five clusters that did not contain a sufficient number of human-labeled ROIs to be given a cell type label (<3 cells in these clusters had matching human labels; these are labeled 'NEW 1' through 'NEW 5'). To examine which neurons these might correspond to, we examined the high-confidence AutoCellLabeler labels for ROIs in these clusters. This produced enough labels to categorize four of these five clusters as SAAD, SMBD, VB02, and VB02. The repeated VB02 label is likely an indication of under-clustering (i.e. the two VB02 clusters should have been merged into the same cluster). The identity of the fifth cluster was unclear, as the ROIs in that cluster were not well labeled by either humans or AutoCellLabeler.

Finally, we examined whether CellDiscoveryNet was able to label cells not detected via AutoCell-Labeler. Specifically, we determined the fraction of the cells detected by CellDiscoveryNet that were labeled by AutoCellLabeler, which was 86%. The new unsupervised detections (the remaining 14%) included: new labels for cells that were otherwise well-labeled by AutoCellLabeler (e.g.: M3); the detection and labeling of several cell types that were uncommonly labeled by AutoCellLabeler (e.g.: RMEV); and the previously mentioned cell type that could not be identified based on human labels. This suggests that the unsupervised approach that we describe here is able to provide cell annotations that were not possible via human labeling or AutoCellLabeler. Overall, these results show how CellDiscoveryNet, together with our clustering approach, can identify >120 cell type classes that occur across animals and assign cells to these classes with 93% accuracy. This unsupervised discovery of cell types occurs in the absence of any human labels.

## Discussion

Aligning and annotating the cell types that make up complex tissues remains a key challenge in computational image analysis. We trained a series of deep neural networks that allow for automated non-rigid registration and neuron identification in the context of brain-wide calcium imaging in freely moving *C. elegans*. This provides an appealing test case for the development of such tools. *C. elegans* movement creates major challenges with tissue deformation, and the animal has >100 defined cell types in its nervous system. We describe BrainAlignNet, which can perform non-rigid registration of the neurons of the *C. elegans* head, allowing for 99.6% accuracy in aligning individual neurons. We further show that this approach readily generalizes to another system, the jellyfish *C. hemisphaerica*. We also describe AutoCellLabeler, which can automatically label >100 neuronal cell types with 98% accuracy, exceeding the performance of individual human labelers by aggregating their knowledge. Finally, CellDiscoveryNet aligns data across animals to perform unsupervised discovery of stereotyped cell types, identifying >100 cell types of the *C. elegans* nervous system from unlabeled data. These tools should be straightforward to generalize to analyses of complex tissues in other species beyond those tested here.

Our newly described network for freely moving worm registration on average aligns neurons with within ~1.5 pixels of each other. Incorporating the network into a full image processing pipeline allows us to link neurons across time with 99.6% accuracy. Training a network to achieve this high performance highlighted a series of general challenges. For example, our attempt to train this network in a fully unsupervised manner (i.e. to simply align two images with no further information) failed. While the resulting networks aligned RFP images of testing data nearly perfectly, it turned out that the image transformations underlying this registration reflected a scrambling of pixels and that the network was not warping the images in the manner that the animal actually bends. Regularization and a semi-supervised training procedure ultimately prevented this failure mode.

Another limitation was that even with the semi-supervised approach, we were only able to train networks to register images from reasonably well-initialized conditions. Specifically, we provided Euler-registered image pairs that were selected to have moderately similar head curvature (although we note that these examples still had fairly dramatic non-rigid deformations; see *Figure 1*). Solving this problem was sufficient to fully align neurons from freely moving *C. elegans* brain-wide calcium imaging, since clustering could effectively be used to link identity across all timepoints even if our image registration only aligned a subset of the image pairs. Our attempts to train a network to register all time points to one another were unsuccessful, although a variety of approaches could conceivably improve upon this moving forward.

Once we had learned how to optimize BrainAlignNet to work in *C. elegans*, we were able to use a similar approach to align neurons in the jellyfish nervous system. Simple modifications to the DDF generation and loss function allowed us to restrict warping to the two-dimensional space of the jellyfish images, while images were padded to maintain similarity to the imaging paradigm of the worm. Overall, the changes that were made were minimal. The results were also extremely similar to those in *C. elegans*. We again found that Euler alignment of image pairs prior to network alignment was critical for accurate warping, suggesting that solving macro deformations before the non-rigid deformations is a generally robust approach. Moreover, we found that it was feasible to align neurons within ~1.5 pixels of one another in jellyfish, just like in *C. elegans*. Having solved this alignment in multiple systems now, we believe it should be straightforward to similarly modify BrainAlignNet for other species as well.

The AutoCellLabeler network that we describe here successfully automated a task that previously required several hours of manual labeling per dataset. It achieves 98% accuracy in cell identification and confidently labels more neurons per dataset than individual human labelers. This performance required a pixel weighting scheme where the network was trained to be especially sensitive to high-confidence labels of neurons that were not ubiquitously labeled by all human labelers. In other words, the network could aggregate knowledge across human labelers and example animals to achieve high performance. While the high performance of AutoCellLabeler is extremely useful from a practical point of view, we note that AutoCellLabeler still cannot label all ROIs in a given image, which would be the highest level of desirable performance. Our analyses above suggest that it is currently bounded by human labeling of training data, which in turn is bounded by our NeuroPAL image quality and the ambiguity of labeling certain neurons in the NeuroPAL strain.

While improvements in human labeling could improve performance of the network, this analysis also highlighted that it would be highly desirable to perform fully unsupervised cell labeling, where the cell types could be inferred and labeled in multispectral images even without any human labeling. To accomplish this, we developed CellDiscoveryNet, which aligns NeuroPAL images across animals. Together with a custom clustering approach, this enabled us to identify 125 neuron classes, labeling them with 93% accuracy in a completely unsupervised manner. This approach could be very useful within the *C. elegans* system, since it is extremely time-consuming to perform human labeling, and it is conceivable that the NeuroPAL labels may change in different genotypes or if the NeuroPAL trans-gene is modified. Beyond *C. elegans*, these unsupervised approaches should be useful, since most tissues in larger animals do not yet have a full catalog of cell types and therefore would greatly benefit from unsupervised discovery. In this spirit, other recent studies have started to develop approaches for unsupervised labeling of imaging data (*Brbić et al., 2022*; *Kim et al., 2022*; *Liu et al., 2023*), although these efforts were not aimed at identifying the full set of cellular subtypes (>100) in individual images, which was the chief objective of CellDiscoveryNet.

These approaches for registering and annotating cells in dense tissues should be straightforward to generalize to other species. For example, variants of BrainAlignNet could be trained to facilitate alignment of tissue sections or to register single-cell resolution imaging data onto a common anatomical reference. Our results suggest that training these networks on subsets of data with labeled feature points, such as cell centroids (i.e. the semi-supervised approach we use here), will facilitate more accurate solutions that, after training, can still be applied to datasets without any labeled feature points. In addition, variants of AutoCellLabeler could be trained on any multi-color cellular imaging data with manual labels. A pixel-wise labeling approach, together with appropriate pixel weighting during training, should be generally useful to build models for automatic cell labeling in a range of different tissues and animals. Finally, models similar to CellDiscoveryNet could be broadly useful to identify previously uncharacterized cell types in many tissues. It is conceivable that hybrid or iterative versions of AutoCellLabeler and CellDiscoveryNet could lead to even higher performance cell type discovery and labeling.

## Materials and methods
### *C. elegans* strains and genetics
All data were collected from 1-day-old adult hermaphrodite *C. elegans* animals raised at 22 C on nematode growth medium (NGM) plates with *E. coli* strain OP50.

For the GCaMP-expressing animals without NeuroPAL, two transgenes were present: (1) *flvIs17: tag-168::NLS-GCaMP7F+NLS-tagRFPt* expressed under a small set of cell-specific promoters: *gcy-28.d, ceh-36, inx-1, mod-1, tph-1(short), gcy-5, gcy-7*; and (2) *flvIs18: tag-168::NLS-mNeptune2.5*. This resulting strain, SWF415, has been previously characterized (*Atanas et al., 2023*).

For the GCaMP-expressing animals with NeuroPAL, two transgenes were present in the strain: (1) *flvIs17*: described above; and (2) *otIs670*: low-brightness NeuroPAL. This resulting strain, named SWF702, has been previously characterized (*Atanas et al., 2023*).

The animals with *eat-4::NLS-GFP* and *tag-168::NLS-GFP* were also previously described (*Atanas et al., 2023*). As is described in the strain list, *tag-168::NLS-mNeptune2.5* was also co-injected with each of these plasmids to generate the two strains: SWF360 (*eat-4::NLS-GFP; tag-168::NLS-mNeptune2.5*) and SWF467 (*tag-168::NLS-GFP; tag-168::NLS-mNeptune2.5*).

We provide here a list of these four strains:

- **SWF415** *flvIs17[tag-168::NLS-GCaMP7F, gcy-28.d::NLS-tag-RFPt, ceh-36:NLS-tag-RFPt, inx-1::tag-RFPt, mod-1::tag-RFPt, tph-1(short)::NLS-tag-RFPt, gcy-5::NLS-tag-RFPt, gcy-7::NLS-tag-RFPt]; flvIs18[tag-168::NLS-mNeptune2.5]; lite-1(ce314); gur-3(ok2245)*
- **SWF702** *flvIs17; otIs670 [low-brightness NeuroPAL]; lite-1(ce314); gur-3(ok2245)*
- **SWF360** *flvEx450[eat-4::NLS-GFP, tag-168::NLS-mNeptune2.5]; lite-1(ce314); gur-3(ok2245)*
- **SWF467** *flvEx451[tag-168::NLS-GFP, tag-168::NLS-mNeptune2.5]; lite-1(ce314); gur-3(ok2245)*

## Jellyfish strains

For jellyfish recordings, we utilized the following strain, which has been previously described (*Weissbourd et al., 2021*):

*Clytia hemisphaerica:* RFamide::NTR-2A-mCherry.

## *C. elegans* microscope and recording conditions

Data used to train and evaluate the models include previously published datasets (*Atanas et al., 2023*; *Pradhan et al., 2025*; *Dag et al., 2023*) and newly collected data. These animals were recorded under similar recording conditions to those described in our previous study (*Atanas et al., 2023*). There were two types of datasets collected, relevant to this study: freely moving GCaMP/TagRFP data and immobilized NeuroPAL data.

Briefly, all neural data (free-moving and NeuroPAL) were acquired on a dual light-path microscope that was previously described (*Atanas et al., 2023*). The light path used to image GCaMP, mNeptune, and the fluorophores in NeuroPAL at single-cell resolution is an Andor spinning disk confocal system built on a Nikon ECLIPSE Ti microscope. Laser light (405 nm, 488 nm, 560 nm, or 637 nm) passes through a 5000 rpm Yokogawa CSU-X1 spinning disk unit (with Borealis upgrade and dual-camera configuration). For imaging, a 40 x water immersion objective (CFI APO LWD 40 X WI 1.15 NA LAMBDA S, Nikon) with an objective piezo (P-726 PIFOC, Physik Instrumente [PI]) was used to image the worm's head (Newport NP0140SG objective piezo was used in some recordings). A dichroic mirror directed light from the specimen to two separate sCMOS cameras (Zyla 4.2 PLUS sCMOS, Andor) with in-line emission filters (525/50 for GCaMP/GFP, and 570 longpass for tagRFP/mNeptune in freely moving recordings; NeuroPAL filters described below). The volume rate of acquisition was 1.7 Hz (1.4 Hz for the datasets acquired with the Newport piezo).

For recordings, L4 worms were picked 18–22 hr before the imaging experiment to a new NGM agar plate seeded with OP50 to ensure that we recorded 1-day-old adults. Animals were recorded on a thin, flat NGM agar pad (2.5 cm x 1.8 cm x 0.8 mm). On the four corners of this agar pad, we pipetted a single layer of microbeads with diameters of 80 μm to alleviate the pressure of the cover-slip (#1.5) on the worm. Animals were transferred to the agar pad in a drop of M9, after which the coverslip was added.

For NeuroPAL data collection, animals were immobilized via cooling to 1 °C, after which multispectral information was captured. We obtained a series of images from each recorded animal while the animal was immobilized (this has been previously described *Atanas et al., 2023*):

(1-3) Isolated images of mTagBFP2, CyOFP1, and mNeptune2.5: We excited CyOFP1 with a 488 nm laser at 32% intensity using a 585/40 bandpass filter. mNeptune2.5 was then recorded with a 637 nm laser at 48% intensity under a 655LP-TRF filter (to not contaminate this recording with TagRFP-T emissions). mTagBFP2 was then isolated with a 405 nm laser at 27% intensity and a 447/60 bandpass filter.

An image with TagRFP-T, CyOFP1, and mNeptune2.5 (all 'red' markers) in one channel, and GCaMP7f in another channel. As described in our previous study, this image was used for neuronal segmentation and registration to both the freely moving recording and individually isolated marker images. We excited TagRFP-T and mNeptune2.5 with a 561 nm laser at 15% intensity and CyOFP1 and GCaMP7f with a 488 nm laser at 17% intensity. TagRFP-T, mNeptune2.5, and CyOFP1 were imaged using a 570LP filter, and GCaMP7f was data collection used a 525/50 bandpass filter.

Each of these images was recorded for 60 time points. We functionally increased the SNR for each of the images by registering all 60 time points for a given image to one another and averaging the transformed images, creating an average image. To facilitate manual labeling of these datasets, we made a composite, three-dimensional RGB image where we set the mTagBFP2 image to blue, CyOFP1 image to green, and mNeptune2.5 image to red as done by *Yemini et al., 2021*. We manually adjusted channel intensities to optimally match their manual.

### *C. hemisphaerica* microscope and recording conditions

*Clytia* medusae were mounted on a glass slide in artificial seawater (ASW *Weissbourd et al., 2021*) and gently compressed under a glass coverslip, using a small amount of Vaseline as a spacer, so that swimming and margin folding motions were visible but limited. The imaging ROI included a segment of the nerve rings and subumbrella. Widefield epifluorescent images of transgenic mCherry fluorescence were acquired at 20 Hz, with 5 ms exposure times, using an Olympus BX51WI microscope, a photometrics Prime95B camera, and Olympus Cellsens software. Videos without visible motion, or with significant lateral drift, were discarded.

### Availability of data, code, and materials

All data are freely and publicly available at the following link:

- https://doi.org/10.7910/DVN/8UE0H9

All code is freely and publicly available (use main/master branches):

- BrainAlignNet for worms: https://github.com/flavell-lab/BrainAlignNet, *flavell-lab, 2024a* and https://github.com/flavell-lab/DeepReg, *flavell-lab, 2024b*.
- BrainAlignNet for jellyfish: https://github.com/flavell-lab/BrainAlignNet_Jellyfish, *flavell-lab, 2025a*.
- GPU-accelerated Euler registration: https://github.com/flavell-lab/euler_gpu, *flavell-lab, 2024c*.
- ANTSUN 2.0: https://github.com/flavell-lab/ANTSUN, *flavell-lab, 2024d* branch v2.1.0; see also https://github.com/flavell-lab/flv-c-setup, and https://github.com/flavell-lab/FlavellPkg.jl/blob/master/src/ANTSUN.jl, for auxiliary package installation.
- AutoCellLabeler: https://github.com/flavell-lab/pytorch-3dunet, *flavell-lab, 2020* and https://github.com/flavell-lab/AutoCellLabeler, *flavell-lab, 2024g*.
- CellDiscoveryNet: https://github.com/flavell-lab/DeepReg, *flavell-lab, 2024b*.
- ANTSUN 2 U: https://github.com/flavell-lab/ANTSUN-Unsupervised, *flavell-lab, 2024h*.

All materials (strains, etc) are freely available upon request.

### BrainAlignNet

Network architecture

BrainAlignNet's architecture is derived from the DeepReg software package, which uses a variation of a 3-D U-Net architecture termed a LocalNet (*Fu et al., 2020*; *Hu et al., 2018*). BrainAlignNet first has a concatenation layer that concatenates the moving and fixed images together along a new, channel dimension. The resulting 284×120×64×2 image (*x*, *y*, *z*, channel) is then passed as input to the LocalNet, which outputs a 284×120×64×3 dense displacement field (DDF). The DDF defines a coordinate transformation from fixed image coordinates to moving image coordinates, relative to the fixed image coordinate system. So, for instance, if $DDF[x, y, z] = (\Delta x, \Delta y, \Delta z)$, it means that the coordinates (*x,y,z*) in the fixed image are mapped to the coordinates $(x + \Delta x, y + \Delta y, z + \Delta z)$ in the moving image. The network has a final warping layer that applies the DDF to transform the moving image into a predicted fixed image whose pixel at location (*x,y,z*) contains the moving image pixel at location $(x, y, z) + DDF[x, y, z]$. It also has another final warping layer that transforms the fixed image centroids

($x,y,z$) into predicted moving image centroids $(x, y, z) + DDF\,[x, y, z]$. The network's loss function causes it to seek to minimize the difference between its predictions and the corresponding input data.

The LocalNet is at its core a 3-D U-Net with an additional output layer that receives inputs from multiple output levels. In more detail, it has three input levels and three output levels, with $16 \cdot 2^i$ feature channels at the $i$th level for $i \in \{0, 1, 2\}$. It contains an encoder block mapping the input to level 0, followed by two more encoder blocks mapping input level $i$ to level $i + 1$ for $i \in \{0, 1\}$. Each of these three encoder blocks contains a convolutional block, a residual convolutional block, and a 2×2×2 max-pool layer. The convolutional block consists of a 3-D convolutional layer with kernel size 3 that doubles the number of feature channels, followed by a batch normalization layer, followed by a ReLU activation function. The residual convolutional block consists of two convolutional blocks in sequence, except that the input (to the residual convolutional block) is added to the output of the second convolutional block right before its ReLU activation function. The bottom block comes after the encoder block at level 2, mapping input level 2 to output level 2. It has the same architecture as a single convolutional block; notably, it does not contain the max-pool layer.

There are three decoder blocks receiving inputs from the three encoder blocks described above. The first two decoder blocks map output level $i + 1$ to output level $i$ for $i \in \{1, 0\}$; the third one maps output level 0 to the preliminary output with the same $(x, y, z)$ dimensions as the input. Each decoding block consists of an upsampling block, a skip-connection layer, a convolutional block, and a residual convolutional block. The upsampling block contains a transposed 3D convolutional layer with kernel size 3 that halves the number of feature channels and an image resizing layer (run independently on the upsampling block's input) using bilinear interpolation to double each dimension of the image. The output of the resizing layer is then split into two equal pieces along the channel axis and summed, and then added to the output of the transposed convolutional layer. The skip-connection layer appends the output of the mirrored encoder block $i$ (for the third decoder block, this corresponds to the first encoder block) right before that encoder block's max pool layer. The skip-connection layer appends this output to the channel dimension, doubling its size. The convolutional and residual convolutional blocks are identical to those in the encoding block, except that the convolutional block halves the number of input channels instead of doubling it.

Finally, there is the output layer. It takes as input the output of the bottom block, as well as the output of every decoder block. To each of these inputs, it applies a 3D convolutional layer that outputs exactly three channels, followed by an upsampling layer that uses bilinear interpolation to increase the dimensions to the size of the original input images. It then averages together all of these images to compute the final 284×120×64×3 DDF.

## Preprocessing

To train and validate a registration network that aligns neurons across time series in freely moving *C. elegans*, we took several steps to prepare the calcium imaging datasets with images and their corresponding centroids. The preprocessing procedure consisted of (i) selecting two different time points from a single video (fixed and moving time points) at which to obtain RFP images (all images given to the network are from the red channel, which contains the signal from NLS-TagRFP) and neuron centroids; (ii) cropping all RFP images to a consistent size; (iii) performing Euler registration (translation and rotation) to align neurons from the image at the moving time point (moving image) to the image at the fixed time point (fixed image); (iv) creating image centroids for the network, which consist of matched lists of centroid positions of all the neurons in both the fixed and moving images.

## Selection of registration problems

We refer to the task of solving the transformation function that aligns neurons from the moving image to the fixed image as a registration problem. We selected our registration problems based on previously constructed (*Atanas et al., 2023*) image registration graphs using ANTSUN 1.4. In these registration graphs, the time points of a single calcium imaging recording served as vertices. An edge between two time points indicates a registration problem that we will attempt to solve. Edges were preferentially created between time points with higher worm posture similarities.

In ANTSUN 1.4, we selected approximately 13,000 pairs of time points (fixed and moving) per video that had sufficiently high worm posture similarity. These registration problems were solved by

gradient descent using our old image processing pipeline, and ANTSUN clustering yielded linked neuron ROIs across frames that were the basis of constructing calcium traces (*Atanas et al., 2023*). To train BrainAlignNet here, we randomly sampled about 100 problems across a total of 57 animals, ultimately compiling 5176 registration problems for training (some registration problems were discarded during subsequent preprocessing steps). To prepare the validation datasets, we sampled 1466 problems across 22 animals. Testing data was 447 problems from five animals.

## Cropping

The registration network requires all 3D image volumes in training, validation, and testing to be of the same size. Therefore, a crucial step in preprocessing was to crop or pad the images along the *x*, *y*, *z* dimensions to a consistent size of (284, 120, 64). Before reshaping the images, we first subtracted the median pixel value from each image (both fixed and moving) and set the negative pixels to zero. Then, we either cropped or padded with zeros around the centers of mass of these images to make the *x* dimension 284, the *y* dimension 120, and the *z* dimension 64.

## Euler registration

Through experimentation with various settings of the network, we have found that it is difficult for the network to learn large rotations and translations at the same time as smaller nonlinear deformations. Euler registration is far more computationally tractable than nonlinear deformation, so we solved Euler registration for the images before providing them to the network. In Euler registration, we rotate or translate the moving images by a certain amount to create predicted fixed images, aiming to maximize their normalized cross-correlation (NCC) with the fixed image. The NCC between a fixed image $F$ and a predicted fixed image $P$ is defined as follows:

$$NCC = \frac{\sum_{x=0}^{X-1} \sum_{y=0}^{Y-1} \sum_{z=0}^{Z-1} \left( F\left[x, y, z\right] - \mu_F \right) \left( P\left[x, y, z\right] - \mu_P \right)}{XYZ\sqrt{\sigma_F^2 \sigma_P^2}}$$

here $(X, Y, Z)$ are the dimensions of the images, $\mu_F$ is the mean of the fixed image, $\mu_P$ is the mean of the predicted fixed image, $\sigma_F^2$ is the variance of the fixed image, and $\sigma_P^2$ is the variance of the predicted fixed image.

The optimal parameters of translation and rotation that resulted in the highest NCC were determined using a brute-force, GPU-accelerated parameter grid search. To further accelerate the grid search, we projected the fixed and moving images onto the *x-y* plane using a maximum-intensity projection along the *z*-axis (so the above NCC calculation only happens along 2 dimensions). We also downsampled the fixed and moving images by a factor of 4 after the *z* maximal projection. The best parameters identified for transforming the projected images were then applied to each *z*-slice to transform the entire 3D image. Finally, a translation along the *z*-axis (computed via brute force search to minimize the total 3D NCC) was applied. This approach was feasible because the vast majority of worm movement occurs along the *x-y* axes.

## Creating image centroids

We obtained the neuronal ROI images for both the fixed and moving RFP images, designating them as the fixed and moving ROI images, respectively. The full sets of ROIs in each image were obtained using ANTSUN 1.4's image segmentation and watershedding functions. ROI images were then constructed as follows. Each pixel in an ROI image contains an index value: 0 for background, or a positive integer for a neuron. All pixels belonging to a specific neuron have the same index, and pixels belonging to any other neuron have a different index. Since the ROI images are created independently at each time point, their neuronal indices are not a priori consistent across time points. Therefore, we used previous runs of ANTSUN 1.4 to link the ROI identities across time points and generated new ROI images with consistent indices across time points – for example, all pixels with value 6 in one time point correspond to the same neuron as pixels with value 6 in any other time point. We deleted any ROIs with indices that were not present in both the moving and fixed images.

We then cropped these ROI images to the same size and subjected them to Euler transformations using the same parameters as their corresponding fixed and moving RFP images. Next, we computed the centroids of each neuron index in the resulting moving and fixed ROI images. The centroid was defined to be the mean *x*, *y*, and *z* coordinates of all pixels of a given ROI. We stored these centroids

as two lists of equal length (typically, around 110). Note that these lists are now the matched positions of neurons in the fixed and moving images.

Since the network expects image centroids to be of the same size, all neuronal centroids in the fixed and moving images were padded and aggregated into arrays of shape (200, 3), ensuring the same ordering of neurons. The extra entries that do not contain neurons are filled with (−1, −1, −1) to make the total number of neurons equal to 200. We designate the neuronal centroid positions in the fixed and moving ROI images as fixed and moving centroids, respectively.

## Loss functions

Our main custom modifications to the DeepReg network focus on the design of the loss function. In particular, we implemented a new supervised centroid alignment loss component and new regularization loss sub-components. Overall, the loss function consists of three major components:

- **Image loss** $L_I$ captures the difference between the warped moving image and the ground-truth fixed image.
- **Centroid alignment loss** $L_C$ is a supervised portion of the loss function. Given pre-labeled centroids corresponding to ground-truth information about neuron positions in the fixed and moving images, this loss component captures the difference between the predicted moving centroids and the ground-truth moving centroids.
- **Regularization loss** $L_R$ captures the prior that the "simplest" DDF that achieves the desired transform outcome is the best. For example, it's implausible that a pair of neurons that start close together end up on opposite sides of the worm, so a DDF that generates such a transformation would have a high value of regularization loss.

The total loss is then computed as $Loss = w_I L_i + w_C L_C + w_R L_R$. We set $w_I = 1$, $w_C = 0.1$, and $w_R = 1$.

## Image loss

The image loss is the negative of the local squared zero-normalized cross-correlation (LNCC) between the fixed and warped moving RFP images. We designate the fixed image as $X_{true}$ and the warped moving image as $X_{pred}$. Define $E(X)$ as a function that computes the discrete expectation of image $X$ within a sliding cube of side length $n$=16:

$$E(X)[x,y,z] = \frac{1}{n^3} \sum_{i=x}^{x+n-1} \sum_{j=y}^{y+n-1} \sum_{k=z}^{z+n-1} X[i,j,k]$$

We then can compute the discrete sliding variance as

$$V(X) = E(X^2) - E(X)^2$$

The image loss (i.e. negative LNCC) is then defined as

$$L_I = -LNCC = \frac{-\left[E(X_{true} \circ X_{pred}) - E(X_{true}) \circ E(X_{pred})\right]^2}{V(X_{true}) \circ V(X_{pred}) + \epsilon}$$

## Centroid alignment loss

The centroid alignment loss is calculated as the negative of the sum of the Euclidean distances between the moving centroids and the network's predicted moving centroids, averaged across the number of centroids available. We designate the ground truth and network predicted centroids as $N \times 3$ matrices $y_{true}$ and $y_{pred}$, respectively, where $N$ is the number of centroids, and the $i$th row of each matrix represents the coordinates of neuron $i$'s centroid. Centroid alignment loss in the overall loss function is then expressed as follows:

$$L_C = \frac{1}{N} \sum_{i=0}^{N-1} \sqrt{\sum_{d=0,1,2} \left(y_{true}[i,d] - y_{pred}[i,d]\right)^2}$$

## Regularization loss

Our regularization loss function consists of four terms that seek to penalize DDFs that do not correspond to possible physical motion of the worm. Of these terms, gradient norm is unchanged from its previous implementation in the DeepReg package, while the other three components are our additions:

- **Gradient norm loss** $L_{Grad}$ penalizes transformations for being nonuniform.
- **Difference norm loss** $L_{Diff}$ penalizes transformations for moving pixels too far.
- **Axis difference norm loss** $L_{AxisDiff}$ penalizes transformations for moving pixels too far along the $z$-dimension, which is less plausible than movement along the $x$- and $y$-dimensions in our recordings.
- **Nonrigid penalty loss** $L_{Nonrigid}$ penalizes transformations for being nonrigid (i.e. not translation and rotation). (Note that unlike the gradient norm loss, this loss function will not penalize DDFs that apply rigid-body rotations.)

We then set $L_R = 0.02L_{Grad} + 0.005L_{Diff} + 0.001L_{AxisDiff} + 0.02L_{Nonrigid}$.

### Gradient norm

The gradient norm computes the average gradient of the DDF by summing up the central finite difference of the DDF as the approximation of derivatives along the $x$, $y$, and $z$ axes. Specifically, we first approximate the partial derivatives for $m \in \{0, 1, 2\}$ as follows:

$$\frac{\partial D_m}{\partial x} \approx \frac{1}{2} \left( D\left[2: X, 1: Y-1, 1: Z-1, m\right] - D\left[0: X-2, 1: Y-1, 1: Z-1, m\right] \right)$$

$$\frac{\partial D_m}{\partial y} \approx \frac{1}{2} \left( D\left[1: X-1, 2: Y, 1: Z-1, m\right] - D\left[1: X-1, 0: Y-2, 1: Z-1, m\right] \right)$$

$$\frac{\partial D_m}{\partial z} \approx \frac{1}{2} \left( D\left[1: X-1, 1: Y-1, 2: Z, m\right] - D\left[1: X-1, 1: Y-1, 0: Z-2, m\right] \right)$$

These results are then stacked to obtain $\frac{\partial D}{\partial x}, \frac{\partial D}{\partial y}, \text{and} \frac{\partial D}{\partial z}$. The gradient norm is calculated as the squared sum of these derivatives, averaged across all elements:

$$L_{Grad} = \frac{1}{3(X-2)(Y-2)(Z-2)} \sum_{i=0}^{X-3} \sum_{j=0}^{Y-3} \sum_{k=0}^{Z-3} \sum_{m=0}^{2} \left[ \left(\frac{\partial D}{\partial x}\right)^2 + \left(\frac{\partial D}{\partial y}\right)^2 + \left(\frac{\partial D}{\partial z}\right)^2 \right]_{i,j,k,m}$$

### Difference norm

The difference norm computes the average squared displacement of a pixel under the DDF $D$:

$$L_{Diff} = \frac{1}{3XYZ} \sum_{i=0}^{X-1} \sum_{j=0}^{Y-1} \sum_{k=0}^{Z-1} \sum_{m=0}^{2} \left( D\left[i, j, k, m\right] \right)^2$$

where $X, Y, Z$ are the sizes of the image along the $x$, $y$, and $z$ axes, respectively.

### Axis difference norm

Axis difference norm of the DDF $D$ calculates the average squared displacement of a pixel along the $z$-axis:

$$D_z = D\left[:, :, :, 2\right] \quad L_{AxisDiff} = \frac{1}{XYZ} \sum_{i=0}^{X-1} \sum_{i=0}^{Y-1} \sum_{k=0}^{Z-1} \left( D_z\left[i, j, k\right] \right)^2$$

### Nonrigid penalty

This term penalizes nonrigid transformations of the neurons by utilizing the gradient information of the DDF. Unlike the approach used in computing the gradient norm, where global rotations would have nonzero gradient, here we are interested in penalizing specifically nonrigid transforms. We accomplish this by constructing a reference DDF, denoted as $D_{ref}$, which warps the entire image to the origin: $D_{ref}\left[x, y, z, :\right] = \left[-x, -y, -z\right]$. Then the difference DDF $D_{diff} = D - D_{ref}$ has the property that the magnitude of its gradient is rotation-invariant. We can then compute $\frac{\partial D_{diff}}{\partial x}, \frac{\partial D_{diff}}{\partial y}, \text{and} \frac{\partial D_{diff}}{\partial z}$ as for the gradient norm and define the gradient magnitude:

$$M = \left(\frac{\partial D_{diff}}{\partial x}\right)^2 + \left(\frac{\partial D_{diff}}{\partial y}\right)^2 + \left(\frac{\partial D_{diff}}{\partial z}\right)^2$$

Under any rigid-body transform, $M = 1$. Thus, the nonrigid penalty is calculated as

$$L_{Nonrigid} = \frac{1}{3\,(X-2)\,(Y-2)\,(Z-2)} \sum_{i=0}^{X-3} \sum_{j=0}^{Y-3} \sum_{k=0}^{Z-3} \sum_{m=0}^{2} \left| M + \frac{1}{M} - 2 \right|_{i,j,k,m}$$

In this way, rigid-body transforms will have 0 loss while any nonrigid transform will have a positive loss.

## Data augmentation

During training, input data was subject to augmentation. We used random affine transformations for augmentation. Each transformation was generated by perturbing the corner points of a cube by random amounts and computing the affine transformation resulting in that perturbation. The same transformation was then applied to the moving image, fixed image, moving centroids, and fixed centroids.

## Optimizer

BrainAlignNet was trained using the Adam optimizer with a learning rate of $10^{-4}$.

## Configuration file

The full configuration file we used during network training is available at https://github.com/flavell-lab/BrainAlignNet/tree/main/configs, *flavell-lab, 2024i*.

## Automatic Neuron Tracking System for Unconstrained Nematodes (ANTSUN) 2.0

We integrated BrainAlignNet into our previously described ANTSUN pipeline (*Atanas et al., 2023*; *Pradhan et al., 2025*; also applied in *Dag et al., 2023*). Briefly, the pipeline: (i) performs some image pre-processing such as shear correction and cropping; (ii) segments the images into neuron ROIs via a 3D U-Net; (iii) finds time points where the worm postures are similar; (iv) performs image registration to define a coordinate mapping between these time points; (v) applies that coordinate mapping to the ROIs; (vi) constructs an ROI similarity matrix storing how likely different ROIs are to correspond to the same neuron; (vii) clusters that matrix to extract neuron identity; (viii) maps the linked ROIs onto the GCaMP data to extract neural traces; and (ix) performs some postprocessing such as background subtraction and bleach correction to extract neural traces.

The differences in ANTSUN 2.0 compared with our previously published version of this pipeline, ANTSUN 1.4, are that in ANTSUN 2.0 we use BrainAlignNet to perform image registration rather than the gradient descent-based elastix, and we modified the heuristic function used to construct the ROI similarity matrix. We only replaced the freely moving registration with BrainAlignNet; the immobilized registrations, channel alignment registration, and freely moving to immobilized registration are still performed with elastix. These remaining elastix-based registrations are much less computationally expensive, taking only about 2% of the total computation time of the original ANTSUN 1.4 pipeline. They will also likely be replaced with BrainAlignNet in a future release of ANTSUN, after further diagnostics and controls are run.

The heuristic function used to compute the ROI similarity matrix was updated to add additional terms specific to BrainAlignNet, including regularization and an additional ROI displacement term that serves to implement our prior that ROIs which moved less far in the registration are more likely to be correctly registered. Letting $i$ and $j$ be two different ROIs in our recording at time points $t_i$ (moving) and $t_j$ (fixed), the full expression for the ROI similarity matrix is:

$$M_{ij} = R_{t_i t_j} \frac{1}{1 + w_1 d_i} q_{t_i t_j}^{w_2} r_{ij}^{w_3} e^{-\left(w_4 a_{ij} + w_5 c_{ij} + w_6 n_{t_i t_j}\right)}$$

where:

$R_{t_i t_j}$ is 1 if there exists a registration mapping $t_i$ to $t_j$, and 0 otherwise.

$d_i$ is the displacement of the centroid of ROI $i$ under the DDF registration between $t_i$ and $t_j$.

$q_{t_i t_j}$ is the registration quality, computed as the NCC of warped moving image $t_i$ vs fixed image $t_j$.

$r_{ij}$ is the fractional overlap of warped moving ROI $i$ and fixed ROI $j$ (intersection/max size).

$a_{ij}$ is the absolute difference in marker channel activity (i.e. tagRFP brightness) between ROIs $i$ and $j$, normalized to mean activity at the corresponding timepoints $t_i$ and $t_j$.

$c_{ij}$ is the distance between the centroids of warped moving ROI $i$ and fixed ROI $j$.

$n_{t_i t_j}$ is the (unweighted) nonrigid penalty loss of the DDF registration from $t_i$ to $t_j$.

$w_i$ are weights controlling how important each variable is.

Additionally, the matrix is forced to be symmetrical by setting $M_{ji} = M_{ij}$ whenever $M_{ji} = 0$ and $M_{ij} \neq 0$. It is also sparse since $R_{t_i t_j}$ and $r_{ij}$ are usually 0. Finally, there are two additional hyperparameters in the clustering algorithm, $w_7$ and $w_8$. $w_7$ controls the minimum height the clustering algorithm will reach (effectively, $w_7$ is a cap on how low $M_{ij}$ values can get, or how low the heuristic value can fall before determining that the ROIs are not the same neuron) and $w_8$ controls the acceptable collision fraction (a collision is defined by a cluster containing multiple ROIs from the same timepoint, which should not happen since each neuron should correspond to only one ROI at each time point).

We determined the weights $w_i$ by performing a grid search through 2912 different combinations of weights on three *eat-4::NLS-GFP* datasets. To evaluate the outcome of each combination, we computed the error rate (rate of incorrect neuron linkages) and number of detected neurons. The error rate was computed as previously described by *Atanas et al., 2023*: since the strain *eat-4::NLS-GFP* expresses GFP in some but not all neurons, we can quantify registration errors as instances where a GFP-positive neuron lacked GFP in a time point and vice versa, as these correspond to neuron mismatches. We then selected the combination of parameters that maximizes the number of detected neurons while minimizing the error rate. One *eat-4::NLS-GFP* dataset (the one shown in *Figure 2*) was used as a withheld testing animal to determine this optimal set of parameters. The pan-neuronal GFP and pan-neuronal GCaMP animals were not included in this parameter search.

The values of the parameters we used were:

$$w_1 = 2$$
$$w_2 = 25$$
$$w_3 = 1$$
$$w_4 = 3$$
$$w_5 = 1$$
$$w_6 = 1$$
$$w_7 = 0.0001$$
$$w_8 = 0.05$$

## Computing registration error using sparse GFP strain eat-4::NLS-GFP

As is described in the text, we computed the error rate in neuron alignment by analyzing ANTSUN-extracted fluorescence for the *eat-4::NLS-GFP* strain. This strain has different levels of GFP in different neurons such that some neurons are GFP-positive and others are GFP-negative. Since alignment is done only using information in the red channel, we can then inspect the green signal to determine whether 'traces' are extracted correctly from this strain. Specifically, the intensity of the GFP signal in a given neuron should be constant across time points. We quantified errors in registration by looking for inconsistencies in the GFP signal across time. However, it was important to account for the fact that only certain errors would be detectable. For example, when a trace from a GFP-negative neuron mistakenly included a timepoint from a GFP-positive timepoint, this could be detected. But, it would not be feasible to detect cases where a trace from a GFP-negative neuron mistakenly included a timepoint from a different GFP-negative neuron. As we described in a previous study (*Atanas et al., 2023*), this was quantified by first setting a threshold of $median\,(F) > 1.5$ to call a neuron as GFP-positive. This threshold resulted in $\mathrm{Frac_{GFP}} = 27\%$ of neurons being called as GFP-positive, which approximately matches expectations for the *eat-4* promoter. Then, for each neuron, we quantified the number of time points where the neuron's activity $F$ at that time point differed from its median by more than 1.5, and exactly one of [the neuron's activity at that time point] and [its median activity] was larger than 1.5. These time points represent mismatches, since they correspond to GFP-negative

neurons that were mismatched to GFP-positive neurons or vice versa. We then computed an error rate of $\frac{\text{number of mismatched time points}}{(\text{number of total time points}) \cdot 2 \cdot \text{Frac}_{\text{GFP}} \cdot (1 - \text{Frac}_{\text{GFP}})}$ as an estimate of the mis-registration rate of our pipeline. The $2 \cdot \text{Frac}_{\text{GFP}} \cdot (1 - \text{Frac}_{\text{GFP}})$ term corrects for the fact that mis-registration errors that send GFP-negative to other GFP-negative neurons, or GFP-positive to other GFP-positive neurons, would otherwise not be detected by this analysis.

## BrainAlignNet for jellyfish

### Architecture

The network architecture for jellyfish BrainAlignNet is essentially as described above, with the notable exception that deformations in the $z$ axis are masked out. This was accomplished after the final upsampling layer of the LocalNet which produces a $W \times D \times H \times 3$ DDF. The layer of this DDF which corresponds to $z$ deformations (the last slice of the 3-depth dimension) was zeroed out, to keep pixels contained within their respective $z$ slice. This was necessary as opposed to simply drastically increasing the weight of the Axis Difference Norm term of the hybrid regularization loss because even small shifts out of plane cause pixel values to decrease when interpolated.

### Preprocessing

In preparation for the training, frames were randomly sampled to make up the 'moving images'. However, in contrast to the *C. elegans* network, the 'fixed images' were all the first frame of each dataset. (However, given the network's ability to generalize to new datasets, it seems that taking this step to standardize fixed images was not necessary.) These image pairs were then cropped to a specified size and padded once above and below along the $z$ axis with the minimum value of the image. It was important that the padding value was within range of the dataset, as padding with zeros would result in normalization drastically reducing the dynamic range of the images if the baseline intensity is not close to zero. This procedure results in WxHx3 images, which the network was trained on. Note that the training image pairs were not Euler aligned. At inference time, the data was initialized via Euler Registration, as described above for *C. elegans*. We found training on harder problems (non-Euler-aligned frame pairs) and then performing inference on Euler-initialized pairs improved performance.

### Loss function

The loss function of Jellyfish BrainAlignNet was mostly unaltered from its standard counterpart, with the exceptions of (1) there was no centroid alignment loss and (2) the gradient norm term of the hybrid regularization loss was modified to no longer consider deformations in the $z$ axis. Without this latter change, the warping generated by the network tended to be blurry and spread out, and thus the gradient norm was calculated as follows:

$$L_{Grad} = \frac{1}{3\,(X-2)\,(Y-2)} \sum_{i=0}^{X-3} \sum_{j=0}^{Y-3} \sum_{m=0}^{2} \left[ \left( \frac{\partial D}{\partial x} \right)^2 + \left( \frac{\partial D}{\partial y} \right)^2 \right]_{i,j,m}$$

### Data augmentation

Likewise, data augmentation was very similar to the *C. elegans* network, utilizing affine transformations as described above. However, as an additional step, each time an image pair was present in training, a random number was generated between 0 and 3, and both images were rotated by 90 degrees that many times. This was vital to allow the network to generalize and avoid overfitting.

### Configuration file

The full configuration file used during training of the jellyfish network is available at https://github.com/flavell-lab/private_BrainAlign_jellyfish/tree/main/configs, *flavell-lab, 2025b*.

## AutoCellLabeler

### Human labeling of NeuroPAL datasets

For cell annotation based on NeuroPAL fluorescence, human labelers underwent training based on the NeuroPAL Reference Manual for strain OH15500 (https://www.hobertlab.org/neuropal/) and an internal lab-generated training manual. They practiced labeling on previously labeled datasets

until they achieved a labeling accuracy of 90% or higher for high-confidence neurons (>3). Following protocols outlined in the NeuroPAL Reference Manual, labelers first identified hallmark neurons that are distinct in color and stable in spatial location. Identification proceeded systematically through anatomical regions, beginning with neurons in the anterior pharynx and ganglion, followed by the lateral ganglion, and concluding with the ventral ganglion. For a trained labeler, each dataset took approximately 4–6 hr to label and each dataset was labeled at least twice by the same observer. Some datasets were verified by an additional labeler to increase certainty. Neurons with different labels from different passes or different labelers had the confidence of those labels decremented to 2 or less, depending on the degree of labeling ambiguity.

## Network architecture

AutoCellLabeler uses a 3-D U-Net architecture (*Atanas et al., 2023*; *Wolny et al., 2020*), with input dimensions 4×64×120×284 (fluorophore channel, $z$, $y$, $x$) and output dimensions 185×64×120×284 (label channel, $z$, $y$, $x$). The 3D U-Net has four input levels and four output levels, with $64 \cdot 2^i$ feature channels at the $i$th level for $i \in \{0, 1, 2, 3\}$.

There is an encoder block that maps an input image to the 0th[h] input level, followed by three additional encoder blocks that map input level $i$ to input level $i + 1$ for $i \in \{0, 1, 2\}$. Each encoder block consists of two convolutional blocks followed by a 2×2×2 max pool layer, with the exception of the first encoder layer which does not have the max pool layer. The first convolutional block in each encoder increases the number of channels by a factor of 2, and the second leaves it unchanged.

Each convolutional block consists of a GroupNorm layer with group size 16 (except for the first convolutional layer in the first encoder, which has group size 1), followed by a 3D convolutional layer with kernel size 3 and the appropriate number of input and output channels, followed by a ReLU activation layer.

After the encoder, the 3D U-Net then has three decoder blocks mapping output level $i + 1$ and input level $i$ to output level $i$ for $i \in \{0, 1, 2\}$. Output level 3 is defined to be the same as input level 3. Each decoder layer consists of a 2×2×2 upsampling layer which upsamples output level $i$ via interpolation, followed by a concatenation layer that concatenates it to input level $i - 1$ along the channel axis, followed by two convolutional blocks. The first convolutional block decreases the number of channels by a factor of 2, and the second convolutional block leaves the number of channels unchanged. After the final decoder layer, a 1×1 convolutional layer is applied to increase the number of output channels to the desired 185.

## Training inputs

We trained the AutoCellLabeler network on a set of 81 human-annotated NeuroPAL images, with 10 images withheld for validation and another 11 withheld for testing. Each training dataset contained three components: image, label, and weight. The images were 4×64×120×284, with the first dimension corresponding to channel: we spectrally isolated each of the four fluorescent proteins NLS-mNeptune 2.5, NLS-CyOFP1, NLS-mTagBFP2, and NLS-tagRFP using our previously described imaging setup (*Atanas et al., 2023*), described in detail above. The training images were then created by registering all of the images to the NLS-tagRFP image as described above (i.e. the other three colors were registered to tagRFP), then cropping all of them to 4×64×120×284 dimensions $(z, y, x)$, and then stacking them along the channel axis to be 4×64×120×284 (in the reverse order that they were for the BrainAlignNet). Images were manually rotated so that the head was pointing in the same direction in all datasets, but there was no Euler alignment of the different datasets to one another. (Note that, due to the data augmentations described below, it is likely that this manual rotation had no effect and was an unnecessary step).

To create the labels, we ran our segmentation U-Net on each such image to generate ROIs corresponding to neurons in these images. Humans then manually annotated the images and assigned a label and a confidence to these ROIs. These confidence values ranged from 1 to 5, with 5 being the maximum. For network training, only confidence-1 labels were excluded while all labels from confidence 2 through 5 were included. We then made a list $\ell$ of length 185: the background label, and all 184 labels that were ever assigned in any of the human-annotated images. This list contained all neurons expected to be in the *C. elegans* head with the exceptions of ADFR, AVFR, RMHL, RMHR, and SABD, as these neurons were not labeled in any dataset. The list also contained six other possible

classes corresponding to neurons in the anterior portion of the ventral cord: VA01, VB01, VB02, VD01, DD01, and DB02, as well as the classes 'glia' and 'granule' to denote non-neuronal objects that fluoresce (and might be labeled with an ROI), and the class 'RMH?' as the human labelers were never able to disambiguate whether their 'RMH' labels corresponded to RMHL or RMHR.

Due to a data processing glitch, labels for 2 of the 81 training datasets were imported incorrectly; validation and testing datasets were unaffected. This resulted in those datasets effectively having random labels during training. We are currently re-training all versions of the AutoCellLabeler network and expect their performance to modestly increase once this is rectified.

For each image, the human labels were transformed into matrices $L$ with dimensions 185×64×120×284 via one-hot encoding, so that $L[n, z, y, x]$ denotes whether the pixel at position $(x, y, z)$ has label $\ell[n]$. Specifically, we set $L[n, z, y, x]$ for $n > 0$ to be 1 if the pixel at position $(x, y, z)$ corresponded to an ROI that the human labeled as $\ell[n]$, and 0 otherwise. For example, the fourth element of $\ell$ was I2L (i.e. $\ell[3]$ = I2L), so $L[3, z, y, x]$ would be 1 in the ROI labeled as I2L and 0 everywhere else. The first label (i.e. $n = 0$) corresponded to the background, which was 1 if all other channels were 0, and 0 otherwise.

Finally, we create a weight matrix $W$ of dimensions 64×120×284 (in the code, this matrix has dimensions 185×64×120×284, but the loss function is mathematically equivalent to the version presented here). The entries of $W$ are determined by the following set of rules for weighting each corresponding pixel in the human label matrix $L$:

- $W[z, y, x] = 1$ for all $x, y, z$ with the background label, that is $L[0, z, y, x] = 1$
- $W[z, y, x] = \frac{130}{N(l_r)} f(c_r)$ if there is an ROI at $(x, y, z)$ with label $l_r$ that has confidence $c_r$. Here, $N(l_r)$ is the number of ROIs across all datasets (train, validation, and testing) with the label $l_r$. This makes neurons with fewer labels more heavily weighted in training. Additionally, $f$ is a function that weighs labels based on human confidence score $c_r$, where $c_r \in \{2, 3, 4, 5\}$. Specifically, $f(2) = 50$, $f(3) = 600$, $f(4) = 900$, and $f(5) = 1000$. The number 130 was the maximum number of times that any neuronal label (e.g.: not 'granule' or 'glia') was detected across all of the training datasets.

For the 'no weight' network described in **Figure 5**, all entries of this matrix were set to 1.

## Loss function

The loss function is pixel-wise weighted cross-entropy loss. This is computed as:

$$Loss = \frac{-1}{XYZK} \sum_{n=0}^{K-1} \sum_{x=0}^{X-1} \sum_{y=0}^{Y-1} \sum_{z=0}^{Z-1} W[z, y, x] \, L[n, z, y, x] \, log\left(\frac{e^{P[n,z,y,x]}}{\sum_{m=0}^{K-1} e^{P[m,z,y,x]}}\right)$$

here $(Z, Y, X)$ are the image dimensions (64, 120, 284), $K$ is the number of total labels (i.e. the length of $\ell$), and $(n, x, y, z)$ are indices within label and image dimensions. $W$ and $L$ are as defined above, and $P$ is the prediction (output) of the network. In this way, the network has a lower loss if $P[n, z, y, x]$ is high when $L[n, z, y, x] = 1$ (i.e.: the network got the label right), as then the softmax $log\left(\frac{e^{P[n,z,y,x]}}{\sum_{m=0}^{K-1} e^{P[m,z,y,x]}}\right)$ term will be close to 0 and therefore multiply $L[n, z, y, x]$ by a small (negative) number, resulting in an overall small (positive) loss. The $W[z, y, x]$ term makes it so the network cares more about pixels and labels with high weight – in particular, it cares more about foreground labels $n > 0$ and about higher-confidence and rarer labels.

## Evaluation metric

The evaluation metric is a weighted mean intersection-over-union (IoU) across channels. Let $A$ be the network's argmax label matrix. Specifically, $A[n, z, y, x] = 1$ when $P[n, z, y, x] = max_m P[m, z, y, x]$ and $A[n, z, y, x] = 0$ otherwise. Then the evaluation metric is defined as:

$$MeanIoU \approx \frac{1}{K} \sum_{n=0}^{K-1} \frac{\sum_{x=0}^{X-1} \sum_{y=0}^{Y-1} \sum_{z=0}^{Z-1} W[z, y, x] \cdot L[n, z, y, x] \cdot A[n, z, y, x]}{\sum_{x=0}^{X-1} \sum_{y=0}^{Y-1} \sum_{z=0}^{Z-1} W[z, y, x] \cdot max(L[n, z, y, x], A[n, z, y, x])}$$

In this manner, if the network is always correct, $A = L$, the numerator and denominator will be equal, and the evaluation score will be 1. Similarly, if the network is always wrong, the evaluation score will be 0. (In the code, this metric is slightly different from the version presented here due to additional complexity with the $W$ matrix having a nonuniform extra dimension, but they act very similarly.)

## Optimizer

The network was optimized with the Adam optimizer with a learning rate of $10^{-4}$.

## Data augmentation

The following data augmentations are performed on the training data. One augmentation is generated for each iteration, in the following order. The same augmentation is applied to the image, label, and weight matrices, except that contrast adjustment and noise are not used for the label and weight matrices. Missing pixels are set to the median of the image, or to 0 for the label and weight matrices. Interpolation is linear for the images and nearest neighbors for label and weight. Full parameter settings such as strength or range of each augmentation are given in the parameter file (see below).

**Rotation**. The rotations in the *xy* plane and *yz* plane are much larger than the rotation in the *xz* plane because the worm is oriented to lay roughly along the *x* axis, and the physics of the coverslip are such that it cannot rotate about the *y* axis.

**Translation**. The image is translated.

**Scaling**. The image is scaled.

**Shearing**. The image is sheared.

**B-Spline deformation**. The B-spline augmentation first builds a 3D B-spline warp by placing a small number of control points evenly along the x-axis and adding Gaussian noise to all their parameters, yielding a smooth (piecewise-cubic), random deformation throughout the volume. On top of that, it introduces a 'worm-bend' in the XY plane by randomly displacing the y-coordinates of successive control points within a specified limit, computing corresponding x-shifts to preserve inter-point spacing (so the chain of points forms an arc). Notably, the training images are set up so that the worm is lying along the x-axis, and this augmentation is the first to execute (e.g.: before the worm can be rotated).

**Rotation by multiples of 180 degrees**. The image is rotated 180 degrees about a random axis (x, y, or z).

**Contrast adjustment**. Each channel is adjusted separately.

**Gaussian blur**. Gaussian blur is added to the image, in a gradient along the *z*-axis. The gradient is intended to mimic the optical effect of the image becoming blurrier farther away from the objective.

**Gaussian noise**. Added to the image, with each pixel being sampled independently.

**Poisson noise**. Added to the image, with each pixel being sampled independently.

## 'Less aug' network training

We trained a version of the network with some of our custom augmentations disabled, to see how important they were to the overall performance, compared with the other more standard data augmentations. The specific augmentations that were disabled were:

The second B-Spline deformation focusing on deformations in the *xy* plane

Contrast adjustment

Gaussian blur

## Parameter file

The full parameter files are available at: https://github.com/flavell-lab/pytorch-3dunet/tree/master/AutoCellLabeler_parameters, *flavell-lab, 2024j*.

They include augmentation hyperparameters and various other settings not listed here. There is a different parameter file for each version of the network, though in most cases, the differences are simply the number of input channels. If a user installs the pytorch-3dunet package from that GitHub repository and replaces the paths to the training and validation data with their locations on your

computer, they can train it with the exact settings we used here. Training will require a GPU with at least 48 GB of VRAM. It also currently requires about 10 GB of CPU RAM per training and validation dataset.

## Evaluation

During evaluation, an additional softmax layer is applied to convert network output into probabilities. Let $I$ be the input image and let $P$ be the network's output (after the softmax layer). Then at every pixel $(x, y, z)$, the network's output array $P[n, z, y, x]$ represents the probability that this pixel has label $\ell[n]$.

## ROI image creation

To convert the output into labels, we first ran our previously described neuron segmentation network *Atanas et al., 2023*; *Pradhan et al., 2025* on the tagRFP channel of the NeuroPAL image. Specifically, since this segmentation network was trained on lower-SNR freely moving data, we ran it on a lower-SNR copy of the tagRFP channel. (This copy was one of the 60 images we averaged together to get the higher-SNR image fed to AutoCellLabeler.)

The segmentation network and subsequent watershed post-processing (*Atanas et al., 2023*) were then used to generate a matrix $R$ with dimensions 284×120×6 (same as the original tagRFP image). Each pixel in $R$ contains an index, either 0 for background or a positive integer indicating a specific neuron. The segmentation network and watershed algorithms were designed such that all pixels belonging to a specific neuron have the same index, and pixels belonging to any other neuron have different indices. We define an ROI $R_i = \{(x, y, z) \mid R[x, y, z] = i\}$.

## ROI label assignment

We now wish to use AutoCellLabeler to assign a label to ROI $R_i$. To do this, we first generate a mask matrix $M_i$ with the same dimensions as $R$, defined by:

- $M_i[x, y, z] = 0$ if $R[x, y, z] \neq i$
- $M_i[x, y, z] = 0.01$ if $R[x, y, z] = i$ and there exists $(x', y', z')$ face-adjacent to $(x, y, z)$ such that $R[x', y', z'] \neq i$.
- $M_i[x, y, z] = 1$ otherwise.

Here, the 0.01 entries are provided to the edges of the ROI so as to weight the central pixels of each ROI more heavily when determining the neuron's identity.

Finally, we define a prediction matrix $D$ that allows us to determine the label of each ROI and the corresponding confidence of each label. Letting $V$ be the number of distinct nonzero values in $R$ (i.e.: the number of ROIs) and $K = 185$ be the number of possible labels (as before), we define a $V \times K$ prediction matrix $D$ whose $(i, n)$ th entry represents the probability that ROI $R_i$ has label $n$ as follows:

$$D[i, n] = \frac{\sum_{xyz} M[i, x, y, z] \, P[n, z, y, x]}{\sum_{xyz} M[i, x, y, z]}$$

Here, the sums are taken over all pixels in the image.

Note that because of the additional softmax layer, we have $\sum_n D[i, n] = 1$ for all $i$. From this, we can then define the label index of ROI $R_i$ to be $n_i = \text{argmax}_n D[i, n]$. From this, we can define its label to be $l[n_i]$, and the confidence of that label to be $D[i, n_i]$.

## ROI label postprocessing

After all ROIs are assigned a label, they are sorted by confidence in descending order. The ROIs are iterated through in this order, with each ROI being assigned its most likely label and the set of all assigned labels being tracked. If an ROI $R_i$ has its most likely label $l_i$ already assigned to a different ROI $R_j$, the distance between the centroids of ROIs $R_i$ and $R_j$ is computed. If this distance is small enough, the collision is likely due to over-segmentation by the segmentation U-Net (i.e. ROIs $R_i$ and $R_j$ are actually the same neuron). In this case, they are assigned the same label. Otherwise,

the collision is likely due to a mistake on the part of AutoCellLabeler, and the label for ROI $R_i$ is deleted. (i.e. the higher-confidence label for ROI $R_j$ is kept and the lower-confidence label $R_i$ is discarded.)

Additionally, ROIs are checked for under-segmentation. This rarely happens when the segmentation U-Net incorrectly merges two neurons into the same ROI. This is assessed by checking how many pixels in the ROI $R_i$ have predictions other than the full ROI label index $n_i$. Specifically, we count the number of pixels with $P[n, x, y, z] > 0.75$ within $R_i$ for some $n \neq n_i$. If there exists at least 10 pixels with label $n \neq n_i$, or 20% of the pixels in the ROI are labeled as $n \neq n_i$ in this way, it is plausible that the ROI contains parts of another neuron. In this case, the label for that ROI is deleted.

Most neuron classes in the *C. elegans* brain are bilaterally symmetric and have two distinct cell bodies on the left and right part of the animal. These are genetically identical and therefore have exactly the same shape and color, which can often make it difficult to distinguish between them. For most applications, it is also usually unnecessary to distinguish between them since they typically have nearly identical activity and function. In some cases, AutoCellLabeler can be confident in the neuron class but uncertain about the L/R subclass, assigning a probability of >10% to both L and R subclasses. In this case, we do not assign a specific subclass, instead assigning a label only for the main class with the sum of its confidence for either of the two subclasses. We note that this is only done for the L/R subclass – other neurons can also have D/V subclasses, but these are typically functionally distinct, so we require the network to disambiguate D/V for all neuron classes.

Finally, certain neuron classes were present few times in our manually labeled data, making it more likely for the network to mislabel them due to lack of training data, and simultaneously making it difficult for us to assess its performance on these neuron classes due to the lack of testing data where they were labeled. We deleted any AutoCellLabeler labels corresponding to one of these classes, which were ADF, AFD, AVF, AVG, DB02, DD01, RIF, RIG, RMF, RMH, SAB, SABV, SIAD, SIBD, VA01, and VD01. Additionally, there are other fluorescent cell types in the worm's head. AutoCellLabeler was trained to label them as either 'glia' or 'granule', to avoid mislabeling them as neurons, and any AutoCellLabeler labels of 'glia' or 'granule' were deleted to ensure all of our analyses are based on actual neuron labels.

Altogether, these postprocessing heuristics resulted in deleting network labels for only 6.3% of ROIs with confidence 4 or greater human neuron labels (i.e.: not 'granule' or 'glia').

## CePNEM simulation analysis (Figure 5E)

To assess the performance of our AutoCellLabeler network on the SWF415 strain, we could not compare its labels to human labels since humans do not know how to label neurons in this strain. Therefore, we used functional information about neuron activity patterns to assess accuracy of the network. We used our previously described CePNEM model to do this (*Atanas et al., 2023*). Briefly, CePNEM fits a single neural activity trace to the animal's behavior to extract parameters about how that neuron represents information about the animal's behavior. CePNEM fits a posterior distribution for each parameter, and statistical tests run on that posterior are used to determine encoding of behavior. For example, if nearly all parameter sets in the CePNEM posterior for a given neuron have the property that they predict the neuron's activity is higher when the animal is reversing, then CePNEM would assign a reversal encoding to that neuron.

By doing this analysis in NeuroPAL animals where the identity of each neural trace is known, we have previously created an atlas of neural encoding of behavior (*Atanas et al., 2023*). This atlas revealed a set of neurons that have consistent encodings across animals: AVA, AVE, RIM, and AIB encode reverse locomotion; RIB, AVB, RID, and RME encode forward locomotion; SMDD encodes dorsal head curvature; and SMDV and RIV encode ventral head curvature. Based on this prior knowledge, we decided to quantify the fraction $f$ of labeled neurons with the expected activity-behavior coupling. For example, if AutoCellLabeler labeled 10 neurons as AVA and 7 of them encoded reverse locomotion when fit by CePNEM, this fraction would be 0.7.

However, this fraction is not necessarily an accurate estimate of AutoCellLabeler's accuracy. For example, it might have been possible for AutoCellLabeler to mislabel a neuron as AVA that happened to encode reverse locomotion in that animal, thus making the incorrect label appear accurate. On the other hand, CePNEM is limited by statistical power and can sometimes fail to detect the appropriate encoding. This could make a correct label appear inaccurate.

To correct for these factors, we ran a simulation analysis to try to estimate the fraction $p$ of labels that were correct. To do this, we iterated through every one of AutoCellLabeler's labels that was one of the consistent-encoding neuron classes (i.e. one of the neurons listed above). In each simulation, we assign labels to neurons in the following manner: with probability $p_{sim}$ (i.e. the fraction of labels estimated by our simulation to be correct), the label was reassigned to a random neuron that was given that label by a human in a NeuroPAL animal (at confidence 3 or greater); with probability $1 - p_{sim}$, the label was reassigned to a random neuron in the same (SWF415) animal. In this way, the simulation controls for both of the possible inaccuracies outlined above. Then the fraction $f_{sim}$ of labeled neurons with the expected encoding was computed for each simulation. 1000 simulation trials were run for each value of $p_{sim}$, which ranged from 0 to 100 – the mean and standard deviation of these trials are shown in *Figure 5E*. We then computed the probability $p_{sim}$ for which $f_{sim}$ was in closest agreement to $f$, which was 69% (dashed vertical line). This is our estimate for the ground-truth correct label probability $p$.

## CellDiscoveryNet

### Network architecture

The architecture of CellDiscoveryNet uses the same LocalNet backbone from DeepReg that Brain-AlignNet uses, with the following modifications to the architecture and training procedure:

The input images to CellDiscoveryNet are 284×120×64×4 instead of 284×120×64.

The image concatenation layer in CellDiscoveryNet concatenates the moving and fixed images along the existing channel dimension instead of adding a new channel dimension. Effectively, this means that the output of that layer (and input to the LocalNet backbone) is now 284×120×64×8 instead of 284×120×64×4.

The affine data augmentation procedure was adjusted to first construct a 3D affine transformation, then independently apply that same transformation to each channel in the 4D input images.

In the output warping layer, the DDF is now applied independently to each channel of the moving image to create the predicted fixed image.

### Loss function

The loss function in CellDiscoveryNet is a weighted sum of image loss and regularization loss. As this is an entirely unsupervised learning procedure, label loss is not used.

The image loss component has weight 1 and uses squared channel-averaged NCC instead of LNCC used by BrainAlignNet. The NCC loss is computed as:

$$L_I = \frac{-1}{C} \sum_{c=1}^{C} \frac{\left( \sum_{x=0}^{X-1} \sum_{y=0}^{Y-1} \sum_{z=0}^{Z-1} \left( F\left[x, y, z\right] - \mu_{Fc} \right) \left( P\left[x, y, z\right] - \mu_{Pc} \right) \right)^2}{(XYZ)^2 \, \sigma_{Fc}^2 \sigma_{Pc}^2}$$

where $(X, Y, Z, C)$ are the dimensions of the image, $F$ is the fixed image, $P$ is the predicted fixed image (i.e.: DDF-transformed moving image), $C$ is the number of channels and $\mu_{Fc}$ and $\mu_{Pc}$ are the mean of channel $c$ in the fixed and predicted images respectively, and $\sigma_{Fc}^2$ and $\sigma_{Pc}^2$ are the variance of channel $c$ in the fixed and predicted images, respectively.

(We note that *Figure 6D* displays the usual NCC computed by treating channel as a fourth dimension.)

The regularization loss terms are as before for BrainAlignNet, except with weights 0 for the axis difference norm, 0.05 for the gradient norm, 0.05 for the nonrigid penalty, and 0.0025 for the difference norm.

### Training data

CellDiscoveryNet was trained on 3240 pairs of 284×120×64×4 images, comprising every possible pair combination of 81 distinct images. These were the same 81 images used to train AutoCellLabeler. Each pair consisted of a moving image and a fixed image. Both images were pre-processed by setting the dynamic range of pixel intensities to [0, 1], independently for each channel.

Each moving image was additionally pre-processed by using our previously described GPU-accelerated Euler registration to coarsely align it to the corresponding fixed image. This registration was run on the NLS-tagRFP channel, and the Euler transform fit to that channel was then independently applied to each other channel to generate the full transformed moving image.

There were 45 validation image pairs (from 10 validation images), and 1866 testing image pairs. The testing image pairs added 11 additional images and consisted of all pairs not present in either the training or validation data. (So, for example, a registration problem between an image in the training data and an image in the validation data would count as a testing image pair, since the network never saw that image pair in training or validation.) The split of images in the validation and testing data was identical to that for AutoCellLabeler.

The network was trained for 600 epochs with the Adam optimizer and a learning rate of $10^{-4}$. Full training parameter settings are available at https://github.com/flavell-lab/DeepReg/tree/main/CellDiscoveryNet/train_config.yaml, *flavell-lab, 2024k*.

## ANTSUN 2U

To convert the CellDiscoveryNet registration outputs into neuron labels across animals, we created a modified version of our ANTSUN image processing pipeline. We skipped the pre-processing steps since the images were already pre-processed, used 102 four-channel images instead of 1600 one-channel images, set the registration graph to be the complete graph (except each pair of images is only registered once and not once in each direction), substituted BrainAlignNet with CellDiscoveryNet for the nonrigid registration step, and skipped the trace extraction steps of ANTSUN (stopping after it computed linked neuron IDs).

We also modified the heuristic function in the matrix that was subject to clustering to better account for the nature of this multi-spectral data. Specifically, we removed the marker channel brightness heuristic $a_{ij}$ since brightness of neurons relative to the mean ROI is not likely to be well conserved across different animals. We replaced it with a more problem-specific heuristic: color. Specifically, the color $C_i$ of an ROI $R_i$ was defined as the 4-vector of its brightness in each of the four channels, normalized to the average brightness of that ROI across the four channels. We then define

$$a_{ij} = \sum_{k=1}^{4} \left| C_{ik} - C_{jk} \right|$$

where $i, j$ each indicates an ROI label.

In this way, $a_{ij}$ will be small if the ROIs have similar colors and large if they have different colors. We use this new color-based $a_{ij}$ in the same way in the heuristic function that we used the original brightness-based $a_{ij}$, except that we set its weight $w_4 = 7$.

We did not run hyperparameter search on any of the other weight parameters $w_i$ for this dataset to avoid overfitting to the 102 animals included in it, instead leaving them all at their default values from the original ANTSUN 2.0 pipeline (with the one exception of $w_8$ which we set to 0 here in light of having much fewer animals than we did timepoints). We hypothesize that performance may increase even further upon hyperparameter search, although this would likely require considerably more data for testing. The only exception was that we varied parameter $w_7$, which controls the precision vs recall tradeoff. Larger values of $w_7$ result in more, but less accurate, detected clusters; each cluster corresponds to a single neuron class label. We elected to use a value of $w_7 = 10^{-9}$ for all displayed results; the full tradeoff curve is available in *Figure 6f*.

### Accuracy metric

By construction, clusters in ANTSUN 2 U should correspond to individual neuron classes. To compute its accuracy, we checked whether clusters indeed only correspond to single neuron classes. Let $L(r, a)$ be the function mapping ROI $r$ in animal $a$ to its human label (ignoring L/R), and let $C_i$ be a set of $(r, a)$ values belonging to the same cluster. We can then define $L_i$ to be the set of labels in $C_i$: $L_i = \{L(r, a) \mid (r, a) \in C_i, L(r, a) \neq UNKNOWN\}$. Then let $F_i$ be the most frequent label in $L_i$. We can then define the accuracy of ANTSUN 2 U as follows:

$$Accuracy = \frac{\sum_{i \in S} \sum_{l \in L_i} \delta_{lF_i}}{\sum_{i \in S} |L_i|}$$

Here, $|L_i|$ is the number of elements in the set $L_i$, $S$ is the set of all clusters with $|L_i| > 2$ (which included all but one cluster in our data with $w_7 = 10^{-9}$), and $\delta$ is the Kronecker delta function.

## Acknowledgements

We thank members of the Flavell lab for critical reading of the manuscript. We thank A Jia for assistance with *Clytia* data collection and B Bunch for assistance in developing *Clytia* BrainAlignNet. TSK acknowledges funding from a MathWorks Science Fellowship. KLC is a Shurl and Kay Curci Awardee of the Life Sciences Research Foundation. BW acknowledges funding from the NIH (R01NS141960, R00NS119749); The Searle Scholars Program; The Klingenstein-Simons Fellowship Award; The Picower Institute for Learning and Memory; and The Freedom Together Foundation. SWF acknowledges funding from NIH (NS131457, GM135413, DC020484); NSF (Award #1845663); the McKnight Foundation; Alfred P Sloan Foundation; The Picower Institute for Learning and Memory; Howard Hughes Medical Institute; and The Freedom Together Foundation. SWF is an investigator of the Howard Hughes Medical Institute.

## Additional information

### Funding

| Funder | Grant reference number | Author |
| --- | --- | --- |
| MIT School of Science | Mathworks Fellowship | Talya S Kramer |
| Life Sciences Research Foundation | Fellowship | Karen L Cunningham |
| National Institutes of Health | NS141960 | Brandon Weissbourd |
| National Institutes of Health | NS119749 | Brandon Weissbourd |
| Searle Scholars Program | Award | Brandon Weissbourd |
| Klingenstein-Simons | Fellowship | Brandon Weissbourd |
| Picower Institute for Learning and Memory | | Brandon Weissbourd Steven W Flavell |
| Freedom Together Foundation | | Brandon Weissbourd Steven W Flavell |
| National Institutes of Health | NS131457 | Steven W Flavell |
| National Institutes of Health | GM135413 | Steven W Flavell |
| National Institutes of Health | DC020484 | Steven W Flavell |
| U.S. National Science Foundation | 1845663 | Steven W Flavell |
| McKnight Foundation | Scholars Award | Steven W Flavell |
| Alfred P. Sloan Foundation | Research Scholar | Steven W Flavell |
| Howard Hughes Medical Institute | Investigator Award | Steven W Flavell |

The funders had no role in study design, data collection and interpretation, or the decision to submit the work for publication.

## Author contributions
Adam A Atanas, Conceptualization, Software, Formal analysis, Investigation, Methodology, Writing – original draft, Writing – review and editing; Alicia Kun-Yang Lu, Brian Goodell, Software, Formal analysis, Investigation, Methodology, Writing – review and editing; Jungsoo Kim, Conceptualization, Software, Investigation, Methodology, Writing – review and editing; Saba N Baskoylu, Di Kang, Talya S Kramer, Eric Bueno, Flossie K Wan, Investigation; Karen L Cunningham, Investigation, Writing – review and editing; Brandon Weissbourd, Funding acquisition, Writing – review and editing; Steven W Flavell, Conceptualization, Funding acquisition, Writing – original draft, Writing – review and editing

## Author ORCIDs
Steven W Flavell ⓘ https://orcid.org/0000-0001-9464-1877

Reviewer #1 (Public review): https://doi.org/10.7554/eLife.108159.2.sa1
Reviewer #2 (Public review): https://doi.org/10.7554/eLife.108159.2.sa2
Reviewer #3 (Public review): https://doi.org/10.7554/eLife.108159.2.sa3
Author response https://doi.org/10.7554/eLife.108159.2.sa4

# Additional files

## Supplementary files
MDAR checklist

## Data availability
All data are freely and publicly available at the following link: https://doi.org/10.7910/DVN/8UE0H9. All code is freely and publicly available (use main/master branches): BrainAlignNet for worms (https://github.com/flavell-lab/BrainAlignNet, *flavell-lab, 2024a* and https://github.com/flavell-lab/DeepReg, *flavell-lab, 2024b*); BrainAlignNet for jellyfish (https://github.com/flavell-lab/BrainAlignNet_Jellyfish, *flavell-lab, 2025a*); GPU-accelerated Euler registration (https://github.com/flavell-lab/euler_gpu, *flavell-lab, 2024c*); ANTSUN 2.0 (https://github.com/flavell-lab/ANTSUN, *flavell-lab, 2024d* [branch v2.1.0]; see also https://github.com/flavell-lab/flv-c-setup, *flavell-lab, 2024e* and https://github.com/flavell-lab/FlavellPkg.jl/blob/master/src/ANTSUN.jl, *flavell-lab, 2024f* for auxiliary package installation); AutoCellLabeler (https://github.com/flavell-lab/pytorch-3dunet, *flavell-lab, 2020* and https://github.com/flavell-lab/AutoCellLabeler, *flavell-lab, 2024g*); CellDiscoveryNet (https://github.com/flavell-lab/DeepReg, *flavell-lab, 2024b*); ANTSUN 2U (https://github.com/flavell-lab/ANTSUN-Unsupervised, *flavell-lab, 2024h*).

The following dataset was generated:

| Author(s) | Year | Dataset title | Dataset URL | Database and Identifier |
|---|---|---|---|---|
| Flavell SW | 2026 | Data and Model Checkpoints | https://doi.org/10.7910/DVN/8UE0H9 | Harvard Dataverse, 10.7910/DVN/8UE0H9 |

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
