## [Editor Report · eLife Assessment]

Whole-brain imaging of neuronal activity in freely behaving animals holds great promise for neuroscience, but numerous technical challenges limit its use. In this **important** study, the authors describe a new set of deep learning-based tools to track and identify the activity of head neurons in freely moving nematodes (*C. elegans*) and jellyfish (*Clytia hemisphaerica*). While the tools **convincingly** enable high tracking speed and accuracy in the settings in which the authors have evaluated them, the claim that these tools should be easily generalizable to a wide variety of datasets is incompletely supported.

---

## [Referee Report · Reviewer #1 (Public review)]

In this important study, the authors develop a suite of machine vision tools to identify and align fluorescent neuronal recording images in space and time according to neuron identity and position. The authors provide compelling evidence for the speed and utility of these tools. While such tools have been developed in the past (including by the authors), the key advancement here is the speed and broad utility of these new tools. While prior approaches based on steepest descent worked, they required hundreds of hours of computational time, while the new approaches outlined here are >600-fold faster. The machine vision tools here should be immediately useful to readers specifically interested in whole-brain *C. elegans* data, but also for more general readers who may be interested in using BrainAlignNet for tracking fluorescent neuronal recordings from other systems.

I really enjoyed reading this paper. The authors had several ground truth examples to quantify the accuracy of their algorithms and identified several small caveats users should consider when using these tools. These tools were primarily developed for *C. elegans*, an animal with stereotyped development, but whose neurons can be variably located due to internal motion of the body. The authors provide several examples of how BrainAlignNet reliably tracked these neurons over space and time. Neuron identity is also important to track, and the authors showed how AutoCellLoader can reliably identify neurons based on their fluorescence in the NeuroPAL background. A challenge with NeuroPAL though, is the high expression of several fluorophores, which compromises behavioral fidelity. The authors provide some possible avenues where this problem can be addressed by expressing fewer fluorophores. While using all four channels provided the best performance, only using the tagRFP and CyOFP channels was sufficient for performance that was close to full performance using all 4 NeuroPAL channels. This result indicates that the development of future lines with less fluorophore expression could be sufficient for reliable neuronal identification, which would decrease the genetic load on the animal, but also open other fluorescent channels that could be used for tracking other fluorescent tools/markers. Even though these tools were developed for *C. elegans* specifically, they showed BrainAlignNet can be applied to other organisms as well (in their case, the cnidarian C. hemisphaerica), which broadens the utility of their tools.

Strengths:

(1) The authors have a wealth of ground-truth training data to compare their algorithms against, and provide a variety of metrics to assess how well their new tools perform against hand annotation and/or prior algorithms.

(2) For BrainAlignNet, the authors show how this tool can be applied to other organisms besides *C. elegans*.

(3) The tools are publicly available on GitHub, which includes useful README files and installation guidance.

Weaknesses:

(1) Most of the utility of these algorithms is for *C. elegans* specifically. Testing their algorithms (specifically BrainAlignNet) on more challenging problems, such as whole-brain zebrafish, would have been interesting. This is a very, very minor weakness, though.

(2) The tools are benchmarked against their own prior pipeline, but not against other algorithms written for the same purpose.

(3) Considerable pre-processing was done before implementation. Expanding upon this would improve accessibility of these tools to a wider audience.

---

## [Referee Report · Reviewer #2 (Public review)]

Summary:

The paper introduced the pipeline to analyze brain imaging of freely moving animals: registering deforming tissues and maintaining consistent cell identities over time. The pipeline consists of three neural networks that are built upon existing models: BrainAlignNet for non-rigid registration, AutoCellLabeler for supervised annotation of over 100 neuronal types, and CellDiscoveryNet for unsupervised discovery of cell identities. The ambition of the work is to enable high-throughput and largely automated pipelines for neuron tracking and labeling in deforming nervous systems.

Strengths:

(1) The paper tackles a timely and difficult problem, offering an end-to-end system rather than isolated modules.

(2) The authors report high performance within their dataset, including single-pixel registration accuracy, nearly complete neuron linking over time, and annotation accuracy that exceeds individual human labelers.

(3) Demonstrations across two organisms suggest the methods could be transferable, and the integration of supervised and unsupervised modules is of practical utility.

Weaknesses:

(1) Lack of solid evaluation. Despite strong results on their own data, the work is not benchmarked against existing methods on community datasets, making it hard to evaluate relative performance or generality.

(2) Lack of novelty. All three models do not incorporate state-of-the-art advances from the respective fields. BrainAlignNet does not learn from the latest optical flow literature, relying instead on relatively conventional architectures. AutoCellLabeler does not utilize the advanced medNeXt3D architectures for supervised semantic segmentation. CellDiscoveryNet is presented as unsupervised discovery but relies on standard clustering approaches, with limited evaluation on only a small test set.

(3) Lack of robustness. BrainAlignNet requires dataset-specific training and pre-alignment strategies, limiting its plug-and-play use. AutoCellLabeler depends heavily on raw intensity patterns of neurons, making it brittle to pose changes. By contrast, current state-of-the-art methods incorporate spatial deformation atlases or relative spatial relationships, which provide robustness across poses and imaging conditions. More broadly, the ANTSUN 2.0 system depends on numerous manually tuned weights and thresholds, which reduces reproducibility and generalizability beyond curated conditions.

Evaluation:

To make the evaluation more solid, it would be great for the authors to (1) apply the new method on existing datasets and (2) apply baseline methods on their own datasets. Otherwise, without comparison, it is unclear if the proposed method is better or not. The following papers have public challenging tracking data: https://elifesciences.org/articles/66410, https://elifesciences.org/articles/59187, https://www.nature.com/articles/s41592-023-02096-3.

Methodology:

(1) The model innovations appear incrementally novel relative to existing work. The authors should articulate what is fundamentally different (architectural choices, training objectives, inductive biases) and why those differences matter empirically. Ablations isolating each design choice would help.

(2) The pipeline currently depends on numerous manually set hyperparameters and dataset-specific preprocessing. Please provide principled guidelines (e.g., ranges, default settings, heuristics) and a robustness analysis (sweeps, sensitivity curves) to show how performance varies with these choices across datasets; wherever possible, learn weights from data or replace fixed thresholds with data-driven criteria.

Appraisal:

The authors partially achieve their aims. Within the scope of their dataset, the pipeline demonstrates impressive performance and clear practical value. However, the absence of comparisons with state-of-the-art algorithms such as ZephIR, fDNC, or WormID, combined with small-scale evaluation (e.g., ten test volumes), makes the strength of evidence incomplete. The results support the conclusion that the approach is useful for their lab's workflow, but they do not establish broader robustness or superiority over existing methods.

Impact:

Even though the authors have released code, the pipeline requires heavy pre- and post-processing with numerous manually tuned hyperparameters, which limits its practical applicability to new datasets. Indeed, even within the paper, BrainAlignNet had to be adapted with additional preprocessing to handle the jellyfish data. The broader impact of the work will depend on systematic benchmarking against community datasets and comparison with established methods. As such, readers should view the results as a promising proof of concept rather than a definitive standard for imaging in deformable nervous systems.

---

## [Referee Report · Reviewer #3 (Public review)]

Context:

Tracking cell trajectories in deformable organs, such as the head neurons of freely moving *C. elegans*, is a challenging task due to rapid, non-rigid cellular motion. Similarly, identifying neuron types in the worm brain is difficult because of high inter-individual variability in cell positions.

Summary:

In this study, the authors developed a deep learning-based approach for cell tracking and identification in deformable neuronal images. Several different CNN models were trained to: (1) register image pairs without severe deformation, and then track cells across continuous image sequences using multiple registration results combined with clustering strategies; (2) predict neuron IDs from multicolor-labeled images; and (3) perform clustering across multiple multicolor images to automatically generate neuron IDs.

Strengths:

Directly using raw images for registration and identification simplifies the analysis pipeline, but it is also a challenging task since CNN architectures often struggle to capture spatial relationships between distant cells. Surprisingly, the authors report very high accuracy across all tasks. For example, the tracking of head neurons in freely moving worms reportedly reached 99.6% accuracy, neuron identification achieved 98%, and automatic classification achieved 93% compared to human annotations.

Weaknesses:

(1) The deep networks proposed in this study for registration and neuron identification require dataset-specific training, due to variations in imaging conditions across different laboratories. This, in turn, demands a large amount of manually or semi-manually annotated training data, including cell centroid correspondences and cell identity labels, which reduces the overall practicality and scalability of the method.

(2) The cell tracking accuracy was not rigorously validated, but rather estimated using a biased and coarse approach. Specifically, the accuracy was assessed based on the stability of GFP signals in the eat-4-labeled channel. A tracking error was assumed to occur when the GFP signal switched between eat-4-negative and eat-4-positive at a given time point. However, this estimation is imprecise and only captures a small subset of all potential errors. Although the authors introduced a correction factor to approximate the true error rate, the validity of this correction relies on the assumption that eat-4 neurons are uniformly distributed across the brain - a condition that is unlikely to hold.

(3) Figure S1F demonstrates that the registration network, BrainAlignNet, alone is insufficient to accurately align arbitrary pairs of *C. elegans* head images. The high tracking accuracy reported is largely due to the use of a carefully designed registration sequence, matching only images with similar postures, and an effective clustering algorithm. Although the authors address this point in the Discussion section, the abstract may give the misleading impression that the network itself is solely responsible for the observed accuracy.

(4) The reported accuracy for neuron identification and automatic classification may be misleading, as it was assessed only on a subset of neurons labeled as "high-confidence" by human annotators. Although the authors did not disclose the exact proportion, various descriptions (such as Figure 4f) imply that this subset comprises approximately 60% of all neurons. While excluding uncertain labels is justifiable, the authors highlight the high accuracy achieved on this subset without clearly clarifying that the reported performance pertains only to neurons that are relatively easy to identify. Furthermore, they do not report what fraction of the total neuron population can be accurately identified using their methods-an omission of critical importance for prospective users.

---

## [Author Response]

**Reviewer #1 (Public review):**
In this important study, the authors develop a suite of machine vision tools to identify and align fluorescent neuronal recording images in space and time according to neuron identity and position. The authors provide compelling evidence for the speed and utility of these tools. While such tools have been developed in the past (including by the authors), the key advancement here is the speed and broad utility of these new tools. While prior approaches based on steepest descent worked, they required hundreds of hours of computational time, while the new approaches outlined here are >600-fold faster. The machine vision tools here should be immediately useful to readers specifically interested in whole-brain *C. elegans* data, but also for more general readers who may be interested in using BrainAlignNet for tracking fluorescent neuronal recordings from other systems.I really enjoyed reading this paper. The authors had several ground truth examples to quantify the accuracy of their algorithms and identified several small caveats users should consider when using these tools. These tools were primarily developed for *C. elegans*, an animal with stereotyped development, but whose neurons can be variably located due to internal motion of the body. The authors provide several examples of how BrainAlignNet reliably tracked these neurons over space and time. Neuron identity is also important to track, and the authors showed how AutoCellLoader can reliably identify neurons based on their fluorescence in the NeuroPAL background. A challenge with NeuroPAL though, is the high expression of several fluorophores, which compromises behavioral fidelity. The authors provide some possible avenues where this problem can be addressed by expressing fewer fluorophores. While using all four channels provided the best performance, only using the tagRFP and CyOFP channels was sufficient for performance that was close to full performance using all 4 NeuroPAL channels. This result indicates that the development of future lines with less fluorophore expression could be sufficient for reliable neuronal identification, which would decrease the genetic load on the animal, but also open other fluorescent channels that could be used for tracking other fluorescent tools/markers. Even though these tools were developed for *C. elegans* specifically, they showed BrainAlignNet can be applied to other organisms as well (in their case, the cnidarian C. hemisphaerica), which broadens the utility of their tools.Strengths:(1) The authors have a wealth of ground-truth training data to compare their algorithms against, and provide a variety of metrics to assess how well their new tools perform against hand annotation and/or prior algorithms.(2) For BrainAlignNet, the authors show how this tool can be applied to other organisms besides *C. elegans*.(3) The tools are publicly available on GitHub, which includes useful README files and installation guidance.

We thank the reviewer for noting these strengths of our study.

Weaknesses:(1) Most of the utility of these algorithms is for *C. elegans* specifically. Testing their algorithms (specifically BrainAlignNet) on more challenging problems, such as whole-brain zebrafish, would have been interesting. This is a very, very minor weakness, though.

We appreciate the reviewer’s point that expanding to additional animal models would be valuable. In the study, we have so far tested our approaches on *C. elegans* and Jellyfish. Given that this is considered a ‘very, very minor weakness’ and that it does not directly affect the results or analyses in the paper, we think this might be better to address in future work.

(2) The tools are benchmarked against their own prior pipeline, but not against other algorithms written for the same purpose.

We agree that it would be valuable to benchmark other labs’ software pipelines on our datasets. We note that most papers in this area, which describe those pipelines, provide the same performance metrics that we do (accuracy of neuron identification, tracking accuracy, etc), so a crude, first-order comparison can be obtained by comparing the numbers in the papers. But, we agree that a rigorous head-to-head comparison would require applying these different pipelines to a common dataset. We considered performing these analyses, but we were concerned that using other labs’ software ‘off the shelf’ on our data might not represent those pipelines in their best light when compared to our pipeline that was developed with our data in mind. Data from different microscopy platforms can be surprisingly different and we wouldn’t want to perform an analysis that had this bias. Therefore, we feel that this comparison would be best pursued by all of these labs collaboratively (so that they can each provide input on how to run their software optimally). Indeed, this is an important area for future study. In this spirit, we have been sharing our eat-4::GFP datasets (that permit quantification of tracking accuracy) with other labs looking for additional ways to benchmark their tracking software.

We also note that there are not really any pipelines to directly compare against CellDiscoveryNet, as we are not aware of any other fully unsupervised approach for neuron identification in *C. elegans*.

(3) Considerable pre-processing was done before implementation. Expanding upon this would improve accessibility of these tools to a wider audience.

Indeed, some pre-processing was performed on images before registration and neuron identification -- understanding these nuances can be important. The pre-processing steps are described in the Results section and detailed in the Methods. They are also all available in our open-source software. For BrainAlignNet, the key steps were: (1) selecting image registration problems, (2) cropping, and (3) Euler alignment. Steps (1) and (3) were critically important and are extensively discussed in the Results and Discussion sections of our study (lines 142-144, 218-234, 318-323, 704-712). Step (2) is standard in image processing. For AutoCellLabeler and CellDiscoveryNet, the pre-processing was primarily to align the 4 NeuroPAL color channels to each other (i.e. make sure the blue/red/orange/etc channels for an animal are perfectly aligned). This is also just a standard image processing step to ensure channel alignment. Thus, the more “custom” pre-processing steps were extensively discussed in the study and the more “common” steps are still described in the Methods. The implementation of all steps is available in our open-source software.

**Reviewer #2 (Public review):**
Summary:The paper introduced the pipeline to analyze brain imaging of freely moving animals: registering deforming tissues and maintaining consistent cell identities over time. The pipeline consists of three neural networks that are built upon existing models: BrainAlignNet for non-rigid registration, AutoCellLabeler for supervised annotation of over 100 neuronal types, and CellDiscoveryNet for unsupervised discovery of cell identities. The ambition of the work is to enable high-throughput and largely automated pipelines for neuron tracking and labeling in deforming nervous systems.Strengths:(1) The paper tackles a timely and difficult problem, offering an end-to-end system rather than isolated modules.(2) The authors report high performance within their dataset, including single-pixel registration accuracy, nearly complete neuron linking over time, and annotation accuracy that exceeds individual human labelers.(3) Demonstrations across two organisms suggest the methods could be transferable, and the integration of supervised and unsupervised modules is of practical utility.

We thank the reviewer for noting these strengths of our study.

Weaknesses:(1) Lack of solid evaluation. Despite strong results on their own data, the work is not benchmarked against existing methods on community datasets, making it hard to evaluate relative performance or generality.

We agree that it would be valuable to benchmark many labs’ software pipelines on some common datasets, ideally from several different research labs. We note that most papers in this area, which describe the other pipelines that have been developed, provide the same performance metrics that we do (accuracy of neuron identification, tracking accuracy, etc), so a crude, first-order comparison can be obtained by comparing the numbers in the papers. But, we agree that a rigorous head-to-head comparison would require applying these different pipelines to a common dataset. We considered performing these analyses, but we were concerned that using other labs’ software ‘off the shelf’ and comparing the results to our pipeline (where we have extensive expertise) might bias the performance metrics in favor of our software. Therefore, we feel that this comparison would be best pursued by all of these labs collaboratively (so that they can each provide input on how to run their software optimally). Indeed, this is an important area for future study. In this spirit, we have been sharing our eat-4::GFP datasets (that permit quantification of tracking accuracy) with other labs looking for additional ways to benchmark their tracking software.

We also note that there are not really any pipelines to directly compare against CellDiscoveryNet, as we are not aware of any other fully unsupervised approach for neuron identification in *C. elegans*.

(2) Lack of novelty. All three models do not incorporate state-of-the-art advances from the respective fields. BrainAlignNet does not learn from the latest optical flow literature, relying instead on relatively conventional architectures. AutoCellLabeler does not utilize the advanced medNeXt3D architectures for supervised semantic segmentation. CellDiscoveryNet is presented as unsupervised discovery but relies on standard clustering approaches, with limited evaluation on only a small test set.

We appreciate that the machine learning field moves fast. Our goal was not to invent entirely novel machine learning tools, but rather to apply and optimize tools for a set of challenging, unsolved biological problems. We began with the somewhat simpler architectures described in our study and were largely satisfied with their performance. It is conceivable that newer approaches would perhaps lead to even greater accuracy, flexibility, and/or speed. But, oftentimes, simple or classical solutions can adequately resolve specific challenges in biological image processing.

Regarding CellDiscoveryNet, our claim of *unsupervised* training is precise: CellDiscoveryNet is trained end-to-end only on raw images, with no human annotations, pseudo-labels, external classifiers, or metadata used for training, model selection, or early stopping. The loss is defined entirely from the input data (no label signal). By standard usage in machine learning, this constitutes unsupervised (often termed “self-supervised”) representation learning. Downstream clustering is likewise unsupervised, consuming only image pairs registered by CellDiscoveryNet and neuron segmentations produced by our previously-trained SegmentationNet (which provides no label information).

(3) Lack of robustness. BrainAlignNet requires dataset-specific training and pre-alignment strategies, limiting its plug-and-play use. AutoCellLabeler depends heavily on raw intensity patterns of neurons, making it brittle to pose changes. By contrast, current state-of-the-art methods incorporate spatial deformation atlases or relative spatial relationships, which provide robustness across poses and imaging conditions. More broadly, the ANTSUN 2.0 system depends on numerous manually tuned weights and thresholds, which reduces reproducibility and generalizability beyond curated conditions.

Regarding BrainAlignNet: we agree that we trained on each species’ own data (worm, jellyfish) and we would suggest other labs working on new organisms to do the same based on our current state of knowledge. It would be fantastic if there was an alignment approach that generalized to all possible cases of non-rigid-registration in all animals – an important area for future study. We also agree that pre-alignment was critical in worms and jellyfish, which we discuss extensively in our study (lines 142-144, 318-321, 704-712).

Regarding AutoCellLabeler: the animals were not recorded in any standardized pose and were not aligned to each other beforehand – they were basically in a haphazard mix of poses and we used image augmentation to allow the network to generalize to other poses, as described in our study. It is still possible that AutoCellLabeler is somehow brittle to pose changes (e.g. perhaps extremely curved worms) – while we did not detect this in our analyses, we did not systematically evaluate performance across all possible poses. However, we do note that this network was able to label images taken from freely-moving worms, which by definition exhibit many poses (Figure 5D, lines 500-525); aggregating the network’s performance across freely-moving data points allowed it to nearly match its performance on high-SNR immobilized data. This suggests a degree of robustness of the AutoCellLabeler network to pose changes.

Regarding ANTSUN 2.0: we agree that there are some hyperparameters (described in our study) that affect ANTSUN performance. We agree that it would be worthwhile to fully automate setting these in future iterations of the software.

Evaluation:To make the evaluation more solid, it would be great for the authors to (1) apply the new method on existing datasets and (2) apply baseline methods on their own datasets. Otherwise, without comparison, it is unclear if the proposed method is better or not. The following papers have public challenging tracking data: https://elifesciences.org/articles/66410, https://elifesciences.org/articles/59187, https://www.nature.com/articles/s41592-023-02096-3.

Please see our response to your point (1) under Weaknesses above.

Methodology:(1) The model innovations appear incrementally novel relative to existing work. The authors should articulate what is fundamentally different (architectural choices, training objectives, inductive biases) and why those differences matter empirically. Ablations isolating each design choice would help.

There are other efforts in the literature to solve the neuron tracking and neuron identification problems in *C. elegans* (please see paragraphs 4 and 5 of our Introduction, which are devoted to describing these). However, they are quite different in the approaches that they use, compared to our study. For example, for neuron tracking they use t->t+1 methods, or model neurons as point clouds, etc (a variety of approaches have been tried). For neuron identification, they work on extracted features from images, or use statistical approaches rather than deep neural networks, etc (a variety of approaches have been tried). Our assessment is that each of these diverse approaches has strengths and drawbacks; we agree that a meta-analysis of the design choices used across studies could be valuable.

We also note that there are not really any pipelines to directly compare against CellDiscoveryNet, as we are not aware of any other fully unsupervised approach for neuron identification in *C. elegans*.

(2) The pipeline currently depends on numerous manually set hyperparameters and dataset-specific preprocessing. Please provide principled guidelines (e.g., ranges, default settings, heuristics) and a robustness analysis (sweeps, sensitivity curves) to show how performance varies with these choices across datasets; wherever possible, learn weights from data or replace fixed thresholds with data-driven criteria.

We agree that there are some ANTSUN 2.0 hyperparameters (described in our Methods section) that could affect the quality of neuron tracking. It would be worthwhile to fully automate setting these in future iterations of the software, ensuring that the hyperparameter settings are robust to variation in data/experiments.

Appraisal:The authors partially achieve their aims. Within the scope of their dataset, the pipeline demonstrates impressive performance and clear practical value. However, the absence of comparisons with state-of-the-art algorithms such as ZephIR, fDNC, or WormID, combined with small-scale evaluation (e.g., ten test volumes), makes the strength of evidence incomplete. The results support the conclusion that the approach is useful for their lab's workflow, but they do not establish broader robustness or superiority over existing methods.

We wish to remind the reviewer that we developed BrainAlignNet for use in worms and jellyfish. These two animals have different distributions of neurons and radically different anatomy and movement patterns. Data from the two organisms was collected in different labs (Flavell lab, Weissbourd lab) on different types of microscopes (spinning disk, epifluorescence). We believe that this is a good initial demonstration that the approach has robustness across different settings.

Regarding comparisons to other labs’ *C. elegans* data processing pipelines, we agree that it will be extremely valuable to compare performance on common datasets, ideally collected in multiple different research labs. But we believe this should be performed collaboratively so that all software can be utilized in their best light with input from each lab, as described above. We agree that such a comparison would be very valuable.

Impact:Even though the authors have released code, the pipeline requires heavy pre- and post-processing with numerous manually tuned hyperparameters, which limits its practical applicability to new datasets. Indeed, even within the paper, BrainAlignNet had to be adapted with additional preprocessing to handle the jellyfish data. The broader impact of the work will depend on systematic benchmarking against community datasets and comparison with established methods. As such, readers should view the results as a promising proof of concept rather than a definitive standard for imaging in deformable nervous systems.

Regarding worms vs jellyfish pre-processing: we actually had the exact opposite reaction to that of the reviewer. We were surprised at how similar the pre-processing was for these two very different organisms. In both cases, it was essential to (1) select appropriate registration problems to be solved; and (2) perform initialization with Euler alignment. Provided that these two challenges were solved, BrainAlignNet mostly took care of the rest. This suggests a clear path for researchers who wish to use this approach in another animal. Nevertheless, we also agree with the reviewer’s caution that a totally different use case could require some re-thinking or re-strategizing. For example, the strategy of how to select good registration problems could depend on the form of the animal’s movement.

**Reviewer #3 (Public review):**
Context:Tracking cell trajectories in deformable organs, such as the head neurons of freely moving *C. elegans*, is a challenging task due to rapid, non-rigid cellular motion. Similarly, identifying neuron types in the worm brain is difficult because of high inter-individual variability in cell positions.Summary:In this study, the authors developed a deep learning-based approach for cell tracking and identification in deformable neuronal images. Several different CNN models were trained to: (1) register image pairs without severe deformation, and then track cells across continuous image sequences using multiple registration results combined with clustering strategies; (2) predict neuron IDs from multicolor-labeled images; and (3) perform clustering across multiple multicolor images to automatically generate neuron IDs.Strengths:Directly using raw images for registration and identification simplifies the analysis pipeline, but it is also a challenging task since CNN architectures often struggle to capture spatial relationships between distant cells. Surprisingly, the authors report very high accuracy across all tasks. For example, the tracking of head neurons in freely moving worms reportedly reached 99.6% accuracy, neuron identification achieved 98%, and automatic classification achieved 93% compared to human annotations.

We thank the reviewer for noting these strengths of our study.

Weaknesses:(1) The deep networks proposed in this study for registration and neuron identification require dataset-specific training, due to variations in imaging conditions across different laboratories. This, in turn, demands a large amount of manually or semi-manually annotated training data, including cell centroid correspondences and cell identity labels, which reduces the overall practicality and scalability of the method.

We performed dataset-specific training for image registration and neuron identification, and we would encourage new users to do the same based on our current state of knowledge. This highlights how standardization of whole-brain imaging data across labs is an important issue for our field to address and that, without it, variations in imaging conditions could impact software utility. We refer the reviewer to an excellent study by Sprague et al. (2025) on this topic, which is cited in our study.

However, at the same time, we wish to note that it was actually reasonably straightforward to take the BrainAlignNet approach that we initially developed in *C. elegans* and apply it to jellyfish. Some of the key lessons that we learned in *C. elegans* generalized: in both cases, it was critical to select the right registration problems to solve and to preprocess with Euler registration for good initialization. Provided that those problems were solved, BrainAlignNet could be applied to obtain high-quality registration and trace extraction. Thus, our study provides clear suggestions on how to use these tools across multiple contexts.

(2) The cell tracking accuracy was not rigorously validated, but rather estimated using a biased and coarse approach. Specifically, the accuracy was assessed based on the stability of GFP signals in the eat-4-labeled channel. A tracking error was assumed to occur when the GFP signal switched between eat-4-negative and eat-4-positive at a given time point. However, this estimation is imprecise and only captures a small subset of all potential errors. Although the authors introduced a correction factor to approximate the true error rate, the validity of this correction relies on the assumption that eat-4 neurons are uniformly distributed across the brain - a condition that is unlikely to hold.

We respectfully disagree with this critique. We considered the alternative suggested by the reviewer (in their private comments to the authors) of comparing against a manually annotated dataset. But this annotation would require manually linking ~150 neurons across ~1600 timepoints, which would require humans to manually link neurons across timepoints >200,000 times for a single dataset. These datasets consist of densely packed neurons rapidly deforming over time in all 3 dimensions. Moreover, a single error in linking would propagate across timepoints, so the error tolerance of such annotation would be extremely low. Any such manually labeled dataset would be fraught with errors and should not be trusted. Instead, our approach relies on a simple, accurate assumption: GFP expression in a neuron should be roughly constant over a 16min recording (after bleach correction) and the levels will be different in different neurons when it is sparsely expressed. Because all image alignment is done in the red channel, the pipeline never “peeks” at the GFP until it is finished with neuron alignment and tracking. The eat-4 promoter was chosen for GFP expression because (a) the nuclei labeled by it are scattered across the neuropil in a roughly salt-and-pepper fashion – a mixture of eat-4-positive and eat-4-negative neurons are found throughout the head; and (b) it is in roughly 40% of the neurons, giving very good overall coverage. Our view is that this approach of labeling subsets of neurons with GFP should become the standard in the field for assessing tracking accuracy – it has a simple, accurate premise; is not susceptible to human labeling error; is straightforward to implement; and, since it does not require manual labeling, is easy to scale to multiple datasets. We do note that it could be further strengthened by using multiple strains each with different ‘salt-and-pepper’ GFP expression patterns.

(3) Figure S1F demonstrates that the registration network, BrainAlignNet, alone is insufficient to accurately align arbitrary pairs of *C. elegans* head images. The high tracking accuracy reported is largely due to the use of a carefully designed registration sequence, matching only images with similar postures, and an effective clustering algorithm. Although the authors address this point in the Discussion section, the abstract may give the misleading impression that the network itself is solely responsible for the observed accuracy.

Our tracking accuracy requires (a) a careful selection of registration problems, (b) highly accurate registration of the selected registration problems, and (c) effective clustering. We extensively discussed the importance of the choosing of the registration problems in the Results section (lines 218-234 and 318-321), Discussion section (lines 704-708), and Methods section (955-970 and 1246-1250) of our paper. We also discussed the clustering aspect in the Results section (lines 247-259), Discussion section (lines 708-712), and Methods section (lines 1162-1206). In addition, our abstract states that the BrainAlignNet needs to be “incorporated into an image analysis pipeline,” to inform readers that other aspects of image analysis need to occur (beyond BrainAlignNet) to perform tracking.

(4) The reported accuracy for neuron identification and automatic classification may be misleading, as it was assessed only on a subset of neurons labeled as "high-confidence" by human annotators. Although the authors did not disclose the exact proportion, various descriptions (such as Figure 4f) imply that this subset comprises approximately 60% of all neurons. While excluding uncertain labels is justifiable, the authors highlight the high accuracy achieved on this subset without clearly clarifying that the reported performance pertains only to neurons that are relatively easy to identify. Furthermore, they do not report what fraction of the total neuron population can be accurately identified using their methods-an omission of critical importance for prospective users.

The reviewer raises two points here: (1) whether AutoCellLabeler accuracy is impacted by ease of human labeling; and (2) what fraction of total neurons are identified. We address them one at a time.

Regarding (1), we believe that the reviewer overlooked an important analysis in our study. Indeed, to assess its performance, one can only compare AutoCellLabeler’s output against accurate human labels – there is simply no way around it. However, we noted that AutoCellLabeler was identifying some neurons with high confidence even when humans had low confidence or had not even tried to label the neurons (Fig. 4F). To test whether these were in fact accurate labels, we asked additional human labelers to spend extra time trying to label a random subset of these neurons (they were of course blinded to the AutoCellLabeler label). We then assessed the accuracy of AutoCellLabeler against these new human labels and found that they were highly accurate (Fig. 4H). This suggests that AutoCellLabeler has strong performance even when some human labelers find it challenging to label a neuron. However, we agree that we have not yet been able to quantify AutoCellLabeler performance on the small set of neuron classes that humans are unable to identify across datasets.

Regarding (2), we agree that knowing how many neurons are labeled by AutoCellLabeler is critical. For example, labeling only 3 neurons per animal with 100% accuracy isn’t very helpful. We wish to emphasize that we did not omit this information: we reported the number of neurons labeled for every network that we characterized in the study, alongside the accuracy of those labels (please see Figures 4I, 5A, and 6G; Figure 4I also shows the number of human labels per dataset, which the reviewer requested). We also showed curves depicting the tradeoff between accuracy and number of neurons labeled, which fully captures how we balanced accuracy and number of neurons labeled (Figures 5D and S4A). It sounds like the reviewer also wanted to know the total number of recorded neurons. The typical number of recorded neurons per dataset can also be found in the paper in Fig. 2E.